# VideoScore2: Think Before You Score In Generated Video Evaluation

[1]**Xuan He**[*†]  [2]**Dongfu Jiang**[*†] [3]**Ping Nie**  [4]**Minghao Liu**  [7]**Zhengxuan Jiang**
[2]**Mingyi Su**  [6]**Wentao Ma**  [6]**Junru Lin**  [2]**Chun Ye**  [6]**Yi Lu**  [2]**Keming Wu**
[2]**Benjamin Schneider**  [2]**Quy Duc Do**  [2]**Zhuofeng Li**  [6]**Yiming Jia**  [8]**Yuxuan Zhang**
[10]**Guo Cheng**  [2]**Haozhe Wang**  [5]**Wangchunshu Zhou**  [9]**Qunshu Lin**  [5]**Yuanxing Zhang**
[2,5]**Ge Zhang**  [5]**Wenhao Huang**  [2]**Wenhu Chen**[†]

[1]**University of Illinois Urbana-Champaign**, [2]**University of Waterloo**, [3]**Independent**, [4]**2077AI**, [5]**M-A-P**,  [6]**University of Toronto**,  [7]**Zhejiang University**,  [8]**University of British Columbia**, [9]**Abaka AI**,  [10]**Netmind.AI**

[†]`xuanhe4@illinois.edu {dongfu.jiang, wenhuchen}@uwaterloo.ca`

Reviewed on OpenReview: `https://openreview.net/forum?id=MpkVh4jH44`

## Abstract

Recent advances in text-to-video generation have produced increasingly realistic and diverse content, yet evaluating such videos remains a fundamental challenge due to their multi-faceted nature encompassing visual quality, semantic alignment, and physical consistency. Existing evaluators and reward models have achieved strong scoring performance, but they often rely on aggregate or dimension-wise scores with limited diagnostic evidence, making it difficult to inspect why a generated video succeeds or fails. We present VIDEOSCORE2, a *multi-dimensional*, *interpretable*, and *human-preference-aligned* evaluator that explicitly assesses visual quality, text-to-video alignment, and physical/common-sense consistency, while producing diagnostic reasoning traces for output inspectability. Our model is trained on a large-scale dataset, VIDEOFEEDBACK2, containing 27,168 videos with human scores and diagnostic feedback across three dimensions. We use rationale-augmented supervised fine-tuning as a cold start, followed by reinforcement learning with Group Relative Policy Optimization (GRPO), where reasoning traces serve as auxiliary supervision for structured evaluation and RL initialization. Extensive experiments demonstrate that VIDEOSCORE2 achieves superior performance, with 44.35 (+5.94) point-score accuracy on our in-domain benchmark and 50.37 (+4.32) average performance across four out-of-domain benchmarks. Ablation studies further show that reasoning traces are most useful as warm-start supervision for training a structured evaluator. Finally, we demonstrate the effectiveness of VIDEOSCORE2 as a reward model for Best-of-N sampling, showing its potential to support both video evaluation and controllable generation.

## 1 Introduction

Recent progress in text-to-video (T2V) generation (OpenAI, 2024; Kuaishou, 2025; Wan et al., 2025) has enabled models to produce increasingly realistic and coherent videos, expanding their potential across domains such as entertainment, education, and simulation. Yet, evaluating the quality of such videos remains a core bottleneck. Unlike text or image evaluation, video assessment must jointly consider visual fidelity, semantic adherence to prompts, and physical plausibility. These dimensions require the backbone vision-language

---

* Equal contribution. Xuan He is the project leader.

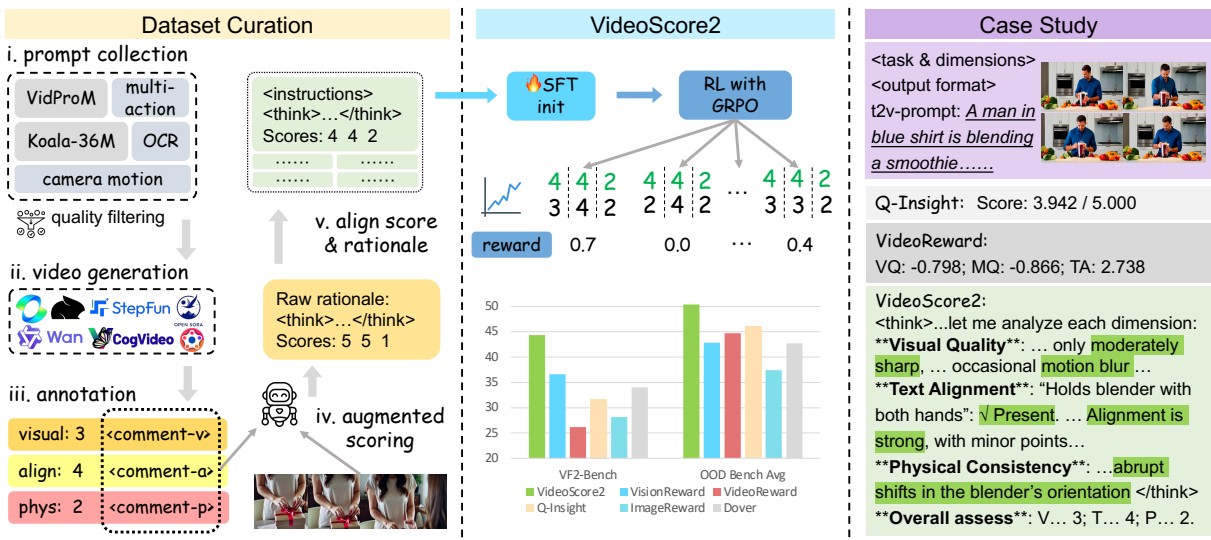

Figure 1: Overview of VIDEOSCORE2. We curate data from five different prompt sources and 22 T2V models with human annotated scores and rationale, which is further used for 2-stage training (cold-start SFT then RL) to elicit the model's thinking ability before scoring.

model (VLM) to possess strong visual understanding ability as well as broad physical and commonsense knowledge.

Many powerful video evaluators/reward models have been developed these days and demonstrate strong performance in various video preference and point-score benchmarks. Examples works include VideoScore (He et al., 2024), VideoPhy2 (Bansal et al., 2025), VideoReward (Liu et al., 2025), etc. In parallel, multi-dimensional preference models such as VisionReward (**?**) show the benefit of decomposing human preferences into finer-grained aspects. However, existing video evaluators are still largely score-centered: they usually produce aggregate or dimension-wise ratings with limited diagnostic evidence, making it difficult to inspect why a video succeeds or fails along different evaluation axes. What's more, these models are all trained via supervised fine-tuning on the collected dataset directly, limiting their generalization ability on the OOD data points (Chu et al., 2025).

To address this gap, we propose VIDEOSCORE2, a *multi-dimensional*, *interpretable*, and *human-aligned* evaluator for AI-generated videos. VIDEOSCORE2 outputs structured scores along three axes—**(1) visual quality**, **(2) text alignment**, and **(3) physical/common-sense consistency**—together with diagnostic reasoning traces that summarize observable evidence behind the ratings. We treat these traces as auxiliary supervision for warm-starting the evaluator, enforcing dimension-aware assessment, and improving the inspectability of model outputs, while the final evaluation quality is validated through human score agreement and downstream benchmarks. Furthermore, compared with SFT-only evaluators, VIDEOSCORE2 combines rationale-augmented cold-start SFT with RL-based alignment and demonstrates strong generalization across diverse OOD benchmarks.

For training VIDEOSCORE2, we curated a large-scale dataset of multi-dimensional evaluations, VIDEOFEED-BACK2, that combines quality scores with reasoning traces. The text-to-video prompts were sourced from both existing datasets and our manually designed cases in some special scenarios (e.g., multiple actions, OCR text, camera motion). The videos were generated by over twenty T2V models spanning from early baselines to recent state-of-the-art modern generative systems, marking a more fine-grained video quality gradient. Our annotators were instructed to provide scores (1–5) as well as brief diagnostic comments across three dimensions, which were later expanded into model-readable reasoning traces through an LLM-assisted semi-blind scoring and alignment pipeline. These expanded traces are used as auxiliary warm-start supervision, while the human-provided scores remain the primary supervision signal for evaluation quality. This design yields a diverse, reliable, and reasoning-augmented dataset that serves as the foundation for teaching

VIDEOSCORE2 both what to evaluate and how to reason. As a result, we derived **2933 unique prompts**, **27168 generated videos** and **81504 scores with rationales** in total. We also split **500** examples as a new video point-score benchmark: VIDEOSCORE-BENCH-V2.

During the experiments, we adopt a two-stage training pipeline. First, a cold-start supervised fine-tuning (SFT) is applied to instill structured output formatting and basic reasoning capabilities. Then, we employ Group Relative Policy Optimization (GRPO) to further align the evaluator model with human preference distributions and improve robustness beyond the SFT initialization. The rationale-augmented cold start provides a structured policy initialization for RL, reducing invalid outputs and encouraging consistent use of the three evaluation dimensions.

Extensive experiments demonstrate the effectiveness of VIDEOSCORE2. On the in-domain test set, VIDEOSCORE2 achieves 44.35 (+5.94) in point-score accuracy, 90.78 (+4.01) in relaxed accuracy and 60.37 (+8.32) in PLCC with significant improvements compared to the previous SoTA. Our model achieves superior performance in the out-of-domain (OOD) benchmarks, reaching 50.37 (+4.32) average performance across 2 OOD preference and 2 OOD point score benchmarks. Furthermore, we show VIDEOSCORE2's potential to be applied as a reward model for T2V generation via Best-of-N (BoN) sampling. We also conduct detailed ablation studies on rationale-augmented SFT, cold-start initialization for RL, and score output format. The results show that reasoning traces are most useful as warm-start supervision for training a structured evaluator, and that combining rationale-augmented cold-start SFT with RL yields the best overall performance.

## 2 Related Works

### 2.1 Text-to-Video Generation

Research on text-to-video (T2V) generation has progressed rapidly with the introduction of large diffusion and Transformer-based architectures. Early milestones include ModelScope (Wang et al., 2023), which provided one of the first open-source diffusion pipelines for T2V, making the task widely accessible. Subsequent VideoCrafter2 (Chen et al., 2024) improved temporal fidelity with enhanced motion realism under data constraints. More recently, CogVideoX (Yang et al., 2024) employed a large DiT backbone to achieve high resolution and narrative coherence. At the industrial scale, OpenAI Sora (OpenAI, 2024) positions itself as a "world simulator," capable of generating long videos with rich physical plausibility. Similarly, an open-sourced work StepVideo-T2V (Ma et al., 2025a) emphasizes scalable training and efficient architecture design to support long and coherent video synthesis. Other commercial systems such as Veo 3 (Google, 2025a), Kling-1.6 (Kuaishou, 2025), and Pika-2.2 (Pika-Labs, 2025) further highlight advances in controllability, and human-centric generation. Despite these achievements, systematic and human-aligned evaluation of video qualities from visual perception to semantic reasoning remains limited, underscoring the need for multi-dimensional and interpretable evaluation frameworks.

### 2.2 Reward Modeling for Vision

Reward modeling has become a central paradigm for aligning generative models with human preferences in both image and video domains. Early methods such as Dover (Wu et al., 2023a), ImageReward (Xu et al., 2023), and HPS-based metrics (Wu et al., 2023c; Ma et al., 2025d) provide scalar preference or quality scores, which are effective but insufficient for capturing the multi-faceted nature of visual quality. More recent approaches—VideoReward (Liu et al., 2025), UnifiedReward (Wang et al., 2025b), VideoPhy2 (Bansal et al., 2025), and VisionReward (**?**)—introduce fine-grained or multi-dimensional scoring for image and video generation, yet are still largely centered on numeric ratings rather than detailed explanatory reasoning. In parallel, recent evaluation benchmarks and feedback datasets have moved beyond coarse preference labels by emphasizing failure-mode diagnosis, explainable assessment, and richer human feedback (Hayes et al., 2025; Hua et al., 2025; Liang et al., 2024). However, they primarily focus on benchmark construction or localized feedback, without providing the long-form analytical reasoning for evaluation. Addressing these limitations, VIDEOSCORE2 delivers multi-dimensional assessments together with long-form analytical reasoning, making its evaluations both human-aligned and interpretable. Moreover, unlike many existing reward models whose

reliance on SFT training often leads to poor generalization, VIDEOSCORE2 demonstrates robust performance across OOD benchmarks, underscoring its potential as a more reliable evaluator.

## 2.3 Video Understanding and Reasoning

Video understanding and reasoning has been a long-standing problem in multimodal learning. Since 2022, transformer-based models have become the backbone of video understanding. Works like Video Swin Transformer (Liu et al., 2021) and InternVideo (Wang et al., 2022) show the benefit of large-scale pretraining and hierarchical temporal modeling. Extending to video–language reasoning, models such as Video-LLaMA (Zhang et al., 2023), Video-LLaVA (Lin et al., 2024b), and mPLUG-Owl-V (Ye et al., 2024) align LMMs with video for open-ended QA and grounding. Recent efforts emphasize long video understanding, including LongVLM (Weng et al., 2024), Video-ChatGPT (Maaz et al., 2024), and benchmarks like Video-MME (Fu et al., 2025) and VideoEval-Pro (Ma et al., 2025b), which stress more realistic, open-ended evaluation of extended video reasoning.

Table 1: Comparison of VIDEOSCORE2 and existing reward models for multi-dimensions, rationale support, and dataset recency.

| Method | Input | Multi-Dim | Rationale | data recency (yy.mm) |
|---|---|---|---|---|
| DeQA-Score | image | ✗ | ✗ | 24.10 |
| Q-Insight | image | ✓ | ✓ | 24.10 |
| VisionReward | video | ✗ | ✗ | 24.08 |
| VideoReward | video | ✓ | ✗ | 24.09 |
| UnifiedReward | video | ✓ | ✗ | 24.12 |
| VIDEOSCORE2 | video | ✓ | ✓ | 25.04 |

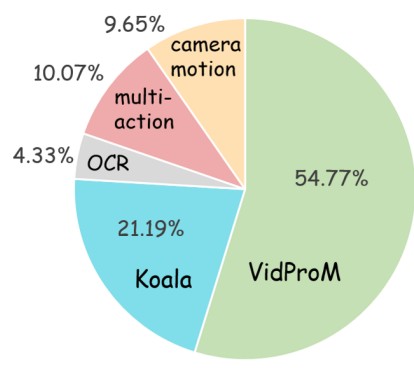

Figure 2: Prompt source proportion.

# 3 Dataset Curation

## 3.1 Data Preparation

**Prompt Collection.** Our dataset prompts come from two sources: **existing datasets** VidProM (Wang & Yang, 2024) and Koala-36M (Wang et al., 2025a), and **manually collected ones**. VidProM provides real user queries from generative model communities, while Koala-36M contains detailed and structured captions that can be adapted into text-to-video prompts. Since some raw prompts are abstract, incomplete or unsuitable, we adopt a two-stage filtering pipeline: (1) **rule-based filtering** is applied to remove prompts that are unsuitable due to length, format, or other constraints; (2) **LLM semantic filtering or revising** is used to discard or revise prompts that are abstract, incoherent, or bad for short video generation. See details in Appendix A.1.

For the manually collected part, we focus on three categories: *multi-action*, *OCR text*, and *camera motion*. This design is motivated by known limitations of current text-to-video (T2V) models in failing to express multiple actions (Wang et al., 2024b), render readable text (Liu et al., 2024), and reproduce camera motion (Hou & Chen, 2025). To construct the multi-action and OCR text prompts, we first design about 100 seed examples and then ask LLMs to expand them creatively, while camera motion prompts are built by appending motion instructions (e.g., "pan left", "tilt up") to some sampled prompts directly. These challenging subsets account for less than 25% of the training prompts, and the remaining prompts are sampled from general-purpose prompts covering diverse scenarios, ensuring the inclusion of former subsets is unlikely to dominate or bias the learned scoring model.

**Video Collection** We collected videos from over 20 T2V models, ranging from early diffusion systems such as ModelScope (Wang et al., 2023) to advanced generators such as StepVideo-T2V (Ma et al., 2025a) and Kling-1.6 (Kuaishou, 2025). For annotation, models were grouped into four coarse tiers (Poor/Early, Medium, Good, Perfect/Modern). For each prompt, we randomly sampled 10 models to generate videos,

Table 2: Definition and Checklist of Evaluation Dimensions.

| Dimension | Definition and Checklist |
|---|---|
| Visual Quality | Quality of visual viewing experience, including resolution, overall and local clarity, smoothness, brightness stability, distortions, etc. |
| Text Alignment | The alignment between video content and text prompt, in terms of subjects, actions, details, styles, and sequential events, etc. |
| Physical / Common-sense Consistency | Whether the video is normal and aligns with common sense, or physical laws, based on everyday knowledge and intuition. Check for abrupt changes, distortions, counterintuitive scenes, and anything weird and abnormal. |

Table 3: Inter-Annotator Agreement (IAA) results ($\mathcal{R}$ / $\alpha$). $\mathcal{R}$ = Relaxed Match (all annotator scores within a margin of 1), $\alpha$ = Krippendorff's Alpha.

| Trial | VQ | TA | PC |
|---|---|---|---|
| 1 ($n = 30$) | 93.33 / 92.06 | 93.33 / 82.71 | 83.33 / 82.99 |
| 2 ($n = 30$) | 96.67 / 90.61 | 80.00 / 77.62 | 80.00 / 80.95 |

Table 4: Human inspection on the difference between model score and human score in augmented-scoring.

| **4951** videos, **14853** scores in total | | | | |
|---|---|---|---|---|
| **Difference** | 0 | 1 | 2 | $\geq 3$ |
| **Counts** | 4710 | 7698 | 2062 | 383 |
| **Bad videos** (diff. $\geq 3$ in any dim.) : **337 / 4951** | | | | |

ensuring a balanced distribution across tiers. This design enabled direct comparisons among videos with the same semantic content but different quality levels, improving scoring consistency and reliability. By covering a wide range of resolutions ($256\times256$ to $1980\times982$), frame rates (8–30 fps), and durations (1–6s), our dataset offers diverse variability, helping VIDEOSCORE2 learn quality from poor output to near-photorealistic generations (see Appendix A.2 for details and A.3 for video examples).

**Evaluation Dimensions** We evaluate videos along three dimensions: **visual quality**, **text alignment**, and **physical / common-sense consistency**, to capture fidelity, semantic accuracy, and content-level reasoning. Unlike VideoScore with five dimensions(He et al., 2024), we remove dynamic degree (mostly prompt-dependent) and subsume temporal consistency under visual quality.

### 3.2 Annotation

We provide dimension-specific checklists in Table 2 to help annotators understand the task. Annotators are required to assign *integer scores* (1–5) and *short comments* for each dimension, later expanded into full rationales by an LLM. For example, the comments can be: "Low resolution, brightness is unstable" or "The second and third actions in prompt are missing". Our team consists of 15 annotators who were trained with annotated examples and pilot rounds (30–50 videos each). Reviewer feedback during these pilot rounds focused on correcting misunderstandings of the annotation guidelines, rather than encouraging agreement on specific videos or enforcing a uniform score distribution. See detailed guidelines in Appendix A.4.

Quality control was ensured through periodic audits, where 10–20% of data was spot-checked for scoring accuracy and comment quality. Annotators with inconsistent work received feedback and were required to revise their annotations. As shown in Table 3, the inter-annotator agreement (IAA) indicates good labeling reliability. Since our human annotation already focuses on compact signals (scores and brief comments), a natural next step is to couple it with active sampling and human-in-the-loop verification, so that only a small, high-impact subset requires full manual attention.

### 3.3 SFT Data Processing

**Rationale Elicitation** We use Claude-4-Sonnet (Anthropic, 2025) (with thinking enabled) to elicit CoT-like rationales (Wei et al., 2023). The LLM receives evaluation instructions, sampled frames, annotator comments (without scores), and 2–3 few-shot examples (see Appendix A.5). We treat the model scores as an auxiliary second rater rather than a source of ground truth. The model's outputs are compared with human scores, if the difference $\leq 1$, human score will be kept; For moderate disagreements (difference= 2),

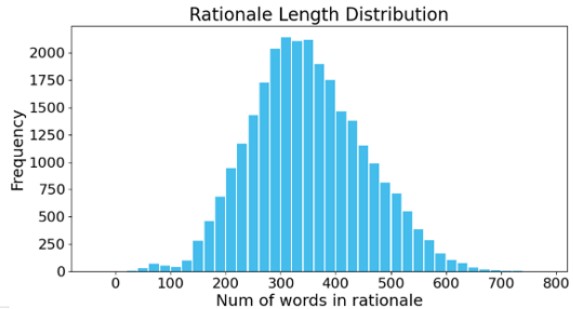

Figure 4: Rationale length (num of words). Most are in 200-600 words.

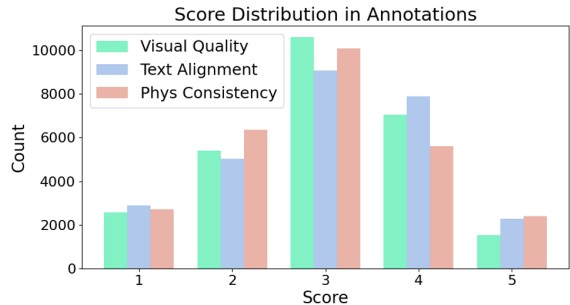

Figure 5: Human annotated score distribution in SFT data.

we assign the midpoint as a soft target to reflect label uncertainty, instead of deterministically overriding the human score; when any dimension differs $\geq 3$, the whole entry is re-scored, up to three times. After resampling, fewer than 10% of entries were discarded. We randomly sample about 20% of each annotator's assigned videosresulting a set of 4,951 videos and report statistics of distribution of human–model differences over flattened dimension scores, i.e., $4,951 \times 3 = 14,853$ scores in Table 4. For example, videos that still contain at least one dimension with difference $\geq 3$ after three attempts are discarded. This criterion discards 337 videos, corresponding to 6.8% of the verified subset and fewer than 10% of the checked entries. Besides, to inspect the quality of expanded rationales, we also conducted human study on randomly sampled items, with full details in Appendix A.4.

**Align Rationales with Scores** Since the final score may occasionally differ from that mentioned in the rationale by one point, we use GPT-5-mini (OpenAI, 2025) to align rationales with scores (prompt template shown in Appendix A.5). This lightweight adjustment preserved the rationale's meaning while ensuring scoring consistency: typically, it involved only minor edits, such as softening or intensifying descriptions of quality issues (e.g., "slight blur" → "noticeable blur"). The rationale length distribution is shown in Figure 4, and the score distribution in Figure 5.

**Building data for SFT and RL** After processing, we obtain 27,168 samples (denoted as VIDEOFEED-BACK2), and the proportions of videos across the four quality tiers (from best to worst) are 10.36%, 33.53%, 41.77%, and 12.54%, respectively (Appendix A.2). 500 videos are held out as the test set (VIDEOSCORE-BENCH-V2) and the rest used for training. The SFT data follow a QA format, where the *query* specifies the task (Table 12), and the *answer* provides rationale and scores. For RL, we follow Video-R1 (Liu et al., 2025), using the same structure with *problem* and *solution* to compute accuracy rewards.

# 4 VideoScore2

## 4.1 Training and Inference Setup

**SFT Cold-Start.** We adopt a two-stage training strategy for VIDEOSCORE2. To ensure basic format-following ability and task familiarity, we first perform supervised fine-tuning (SFT) as the cold-start. The training is implemented with the LLaMA-Factory (Zheng et al., 2024a) framework, and Qwen2.5-VL-7B-Instruct (Bai et al., 2025) as base model.

For preparing the SFT checkpoint to initialize RL, we consider both the performance on VIDEOSCORE-BENCH-V2 and training loss stability: high benchmark scores may indicate overfitting on in-domain tests and weak generalization to others. Balancing these factors, we adopt the configuration described in Appendix D.1 as our main SFT model. Additional ablations on sampling fps, learning rate, and training epochs are reported in Appendix D.2 and D.3.

**Reinforcement Learning** We further train SFT checkpoint with open-source video reinforcement learning framework Video-R1 (Feng et al., 2025) implementing GRPO (Shao et al., 2024) to enhance its analytical robustness and human alignment:

- *Accuracy Reward.* The reward is defined by the degree of match. The design of this reward signal follows the principle that only predictions within $\pm 1$ of the ground truth on all dimensions should receive non-zero reward. On a 5-point scale, a deviation of one point is marginally acceptable, whereas deviations of two or more points indicate serious misjudgments and often contradict the ground-truth evaluation.

$$R_{\mathrm{acc}} = \begin{cases} 1.0 & \text{if all three dimensions match exactly,} \\ 0.7 & \text{if two match and one differs by 1,} \\ 0.4 & \text{if one matches and two differ by 1,} \\ 0.1 & \text{if all three differ by 1,} \\ 0 & \text{otherwise.} \end{cases} \tag{1}$$

- *Format Reward.* To ensure the output includes both a rationale and final scores, we assign $R_{\mathrm{fmt}} = 1$ if the response contains the `<think>` tag with rationale, and $R_{\mathrm{fmt}} = 0$ otherwise.

- *Final Reward.* Following the setting for general video reasoning tasks in Video-R1, final reward $R = R_{\mathrm{acc}} + \lambda R_{\mathrm{fmt}}$.

In practice, when starting from the SFT checkpoint, outputs already follow the query template, making the format reward redundant; thus we set $\lambda = 0$ to focus on accuracy. In contrast, RL from the base model without SFT often shows format deviations, so we set $\lambda = 0.3$ to encourage valid rationales and scores.

RL training uses a learning rate of 2e-6 with $G = 8$ generations per rollout, on 4×A100 GPUs (about 8 hours per 100 steps). Evaluations on VIDEOSCORE-BENCH-V2 and OOD benchmarks show performance peaks at around 300 steps; beyond this, performance on VIDEOSCORE-BENCH-V2 drops (Appendix D.5). We therefore adopt the 300-step checkpoint for all reported results.

**Inference.** Both the SFT and RL models output free-form text with rationale and final scores, following the same query template (Table 12). To generalize discrete predictions $\{1, 2, 3, 4, 5\}$, we set decoding temperature to 0.7 and convert them into soft float scores using token-level probabilities:

$$\tilde{y} = \arg\max_s p(s) \times \frac{p(s)}{\sum_{j=1}^{5} p(j)}. \tag{2}$$

This yields smoother scores in $[1, 5]$ while preserving interpretability. We further ablate score format (int vs. float) in Table 7 and inference fps (2, 4, 8) in Appendix D.4; all reported results use 2 fps and normalized float scores.

### 4.2 Benchmarks

We first evaluate on the *in-domain* VIDEOSCORE-BENCH-V2, which is held out from the same VideoFeedback2 construction pipeline, measuring the performance under the same annotation protocol, score scale, and rationale style as the training data. To further test whether the learned evaluator generalizes beyond dataset-specific annotation patterns, we conduct *out-of-domain* assessment on four external benchmarks with different data sources and annotation pipelines. These benchmarks cover a wide range of quality evaluation scenarios, and can be categorized into two types based on the evaluation task: **Pairwise Preference** and **Point Score**.

**Pairwise preference** benchmarks require the evaluator model to compare a pair of videos and identify which one exhibits higher quality.

- *VideoGenReward-Bench* (Liu et al., 2025), built on VideoGen-Eval (Yang et al., 2025), contains 4,691 videos and 25,234 pairs. Annotators provide pairwise preference labels on dimensions of Visual Quality, Motion Quality, Text Alignment, and Overall preference.

- *T2VQA-DB* (Kou et al., 2024) assigns each video a human quality score (0–100). We sample 2,000 videos and derive 1,822 preference pairs by comparing the scores of videos.

The preference benchmarks both have ties, For models that output float scores, we treat two videos as having equal preference if their score difference is within 5% of the model's score range. (e.g., in $[0.0, 5.0]$, scores $3.28$ vs. $3.26 \Rightarrow$ tie).

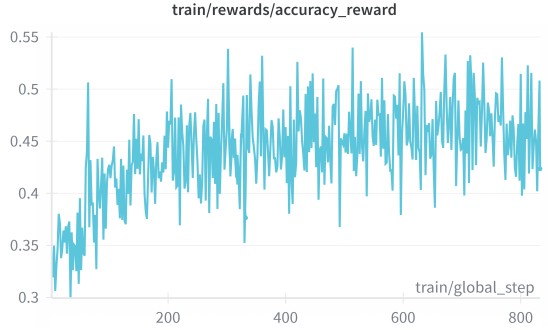
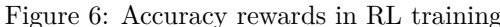

Figure 6: Accuracy rewards in RL training.

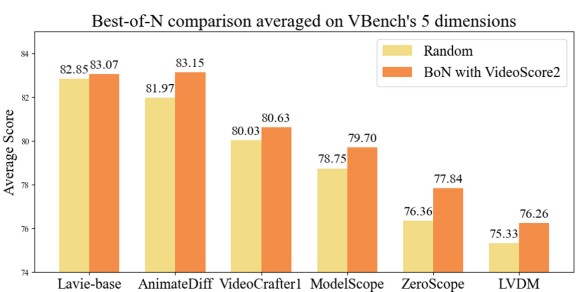

Figure 7: Comparison of Best-of-N sampling with VIDEOSCORE2 and random ones on averaged 5 VBench dimensions.

**Point score** benchmarks focus on how well the evaluator's numerical predictions (after appropriate rescaling) align with the ground-truth scores in overall quality or fine-grained dimensions.

- *MJ-Bench-Video* (Tong et al., 2025) contains 2,170 human-labeled videos with aspect-level scores in {0,1,2}. We use {Fineness, Alignment, Coherence & Consistency} and average VIDEOSCORE2 's three dimensions to compare with their overall score.

- *VideoPhy2-test* (Bansal et al., 2025) includes 3,396 videos annotated on a 1–5 scale for Semantic Adherence and Physical Consistency, which directly match VIDEOSCORE2 's second and third dimensions.

For consistency across benchmarks with different focuses and dimensions, we perform dimension mapping and ground-truth score rescaling, as detailed in Appendix B.1.

### 4.3 Baseline Models

To ensure a comprehensive and rigorous evaluation of VIDEOSCORE2, we evaluate it against more than 10 baseline methods spanning diverse methodological families, divided into two classes.

**Prompting VLMs**, we employ the same query templates as in the training data and provide sampled video frames to current advanced LLMs with vision support for direct scoring, including Gemini-2.5-Pro (Google, 2025b), GPT-5 (OpenAI, 2025), and GLM-4.1v-9b-Thinking (Team et al., 2025). To avoid circular evaluation concern, we exclude Claude-4-Sonnet (Anthropic, 2025), which is used for the rationale expansion. We report both zero-shot and few-shot (3-shot) performances.

**Vision Reward/Scoring Models**, can be categorized based on whether it supports video input.

- *Image-only Models*: We adopt ImageReward (Xu et al., 2023), DeQA-Score (You et al., 2025), and Q-Insight (Li et al., 2025). For video evaluation, frames are sampled at the same fps as the video models. ImageReward and DeQA-Score yield a single overall score, while Q-Insight supports aspect-specific queries, which we align with the three dimensions of VIDEOSCORE2.

- *Video-Capable Models*: We include VideoReward (Liu et al., 2025), UnifiedReward (Wang et al., 2025b), VideoScore (He et al., 2024), and VideoPhy2 (Bansal et al., 2025), which support multi-dimensional scoring. Others such as VisionReward (Xu et al., 2024), Q-Align (Wu et al., 2023b), and DOVER (Wu et al., 2023a) provide only a single overall score.

Most models output scores without detailed reasoning. VisionReward uses fine-grained binary questions aggregated into a score, but lacks explicit explanations. LiFT adds short comments, yet these remain high-level and superficial. In contrast, VIDEOSCORE2 produces both dimension-level scores and comprehensive analyses, making its evaluation more interpretable. Since different models use varying dimensions and scales, we rescale and adjust all outputs to match VIDEOSCORE2 's setting (detailed in Appendix B.2).

Table 5: Accuracy and correlation between model answer and human score on VIDEOSCORE-BENCH-V2. *Relaxed Accuracy* counts cases where the prediction differs from the ground truth by at most one point. **Bold** denotes the best model and the underlined denotes the second best.

| VideoScore-Bench-v2 | Accuracy | | | | Relaxed Accuracy | | | | PLCC | | | |
|---|---|---|---|---|---|---|---|---|---|---|---|---|
| | Visual | Align | Phy | Avg | Visual | Align | Phy | Avg | Visual | Align | Phy | Avg |
| **Prompting VLMs** | | | | | | | | | | | | |
| Gemini-2.5-Pro (0-shot) | 29.92 | 29.72 | 24.07 | 27.90 | 71.49 | 70.88 | 61.45 | 67.94 | 26.71 | 32.96 | 19.75 | 26.47 |
| Gemini-2.5-Pro (3-shot) | 29.03 | 30.44 | 22.58 | 27.35 | 74.60 | 76.82 | 63.71 | 71.71 | 50.24 | 50.65 | 45.66 | 45.66 |
| GPT-5 (0-shot) | 30.72 | 27.91 | 20.08 | 26.24 | 73.90 | 72.29 | 60.84 | 69.01 | 13.38 | 23.34 | 17.24 | 17.99 |
| GPT-5 (3-shot) | 37.90 | 35.28 | 25.61 | 32.93 | 81.25 | 76.41 | 70.16 | 75.94 | 38.27 | 43.51 | 34.39 | 38.72 |
| GLM-4.1v-9B (0-shot) | 33.27 | 31.46 | 21.42 | 28.72 | 80.76 | 77.15 | 61.22 | 73.04 | 28.03 | 16.80 | 10.18 | 18.34 |
| GLM-4.1v-9B (3-shot) | 34.62 | 31.57 | 25.26 | 30.48 | 81.47 | 80.86 | 66.80 | 76.38 | 30.99 | 35.03 | 16.36 | 27.46 |
| **Reward/Scoring Models for Image** | | | | | | | | | | | | |
| ImageReward | 29.06 | 28.06 | 27.26 | 28.13 | 65.13 | 68.94 | 61.72 | 65.26 | 28.23 | 40.76 | 23.26 | 30.75 |
| DeQA-Score | 36.87 | 28.66 | 32.06 | 32.53 | 85.37 | 77.15 | 80.96 | 81.16 | 44.87 | 23.96 | 29.73 | 32.85 |
| Q-Insight | 33.60 | 30.60 | 31.00 | 31.73 | 81.40 | 77.40 | 75.60 | 78.13 | 41.05 | 25.44 | 27.54 | 31.34 |
| **Reward/Scoring Models for Video** | | | | | | | | | | | | |
| VideoScore1.1 | 41.48 | 34.87 | 38.88 | 38.41 | 90.98 | 82.37 | 86.97 | 86.77 | 49.00 | 30.90 | 47.00 | 42.30 |
| VideoReward | 23.45 | 28.86 | - | 26.16 | 60.32 | 67.74 | - | 64.03 | 46.36 | 48.31 | - | 47.34 |
| UnifiedReward | 25.20 | 27.20 | 22.80 | 25.07 | 71.00 | 64.80 | 68.00 | 67.93 | 58.61 | 43.91 | **53.64** | 52.05 |
| VisionReward | 41.28 | 33.47 | 35.07 | 36.61 | 87.17 | 84.37 | 82.16 | 84.57 | 46.85 | 45.32 | 38.25 | 43.47 |
| Q-Align | 28.66 | 28.06 | 27.86 | 28.19 | 75.55 | 69.74 | 68.94 | 71.41 | 54.71 | 34.01 | 37.78 | 42.17 |
| AIGVE-MACS | 20.12 | 12.48 | 14.09 | 15.56 | 62.37 | 46.48 | 45.27 | 51.37 | 27.30 | 6.90 | 13.03 | 15.74 |
| VideoPhy2-AutoEval | - | 28.46 | 16.23 | 22.35 | - | 73.75 | 52.31 | 63.03 | - | 35.42 | 25.41 | 30.42 |
| Dover | 39.08 | 31.06 | 31.86 | 34.00 | 84.77 | 74.75 | 75.92 | 78.48 | 50.24 | 32.83 | 33.00 | 38.69 |
| **VideoScore2** | | | | | | | | | | | | |
| Ours | **50.10** | **43.88** | **39.08** | **44.35** | **92.99** | **91.38** | **87.98** | **90.78** | **60.13** | **62.60** | 52.73 | **60.37** |
| Δ over Best Baseline | +8.62 | +9.01 | +0.20 | +5.94 | +2.80 | +7.01 | +1.01 | +4.01 | +1.52 | +14.29 | -0.91 | +8.32 |

## 4.4 Evaluation Results

We report results on VIDEOSCORE-BENCH-V2 in Table 5, with Accuracy (w/o and w/ relaxation) and correlation metrics (PLCC). For float-output models, scores are rounded for accuracy and kept raw for correlations. VIDEOSCORE2 surpasses the best baseline across all dimensions and metrics. We further test on four out-of-domain (OOD) benchmarks: two pairwise preference and two point-score (Section 4.2). In the tables, *Overall* denotes an explicit overall score, while *Avg* is the mean across dimensions; preference results include ties. As shown in Table 6, while VIDEOSCORE2 is not always the top model on each benchmark, it achieves the highest overall average. The inference cost of our method and the baselines are reported in Table 18 in Appendix C.5.

To further validate the effectiveness of VIDEOSCORE2 in video evaluation, we conduct human inspection to examine whether its predicted scores were reasonable and whether the analyses were accurate and appropriate. Qualitative examples are provided in Appendix E.

## 4.5 Best-of-N Sampling with VideoScore2

We evaluate VIDEOSCORE2 with best-of-$n$ (BoN) sampling ($n = 5$), where the model selects the best video among candidates. Six T2V models of moderate or poor quality are used, avoiding very strong ones to highlight the BoN effect. For 500 prompts, each model generates $500 \times 5$ videos. As the downstream

Table 6: Performance comparison on out-of-domain benchmarks, with 2 pairwise preference benchmarks and 2 point-score benchmarks. **Bold** denotes the best model and the underlined denotes the second best. For "OOD Preference Benchmark," performance is computed over all test samples.

| OOD Bench | Average | OOD Preference Benchmark | | OOD Point Score Benchmark | |
| --- | --- | --- | --- | --- | --- |
| | | VideoGen-Reward Bench | T2VQA-DB (Preference) | MJ-Bench -Video | VideoPhy2 -test |
| Reward/Scoring Models for Image | | | | | |
| ImageReward | 37.40 | 47.14 | 43.46 | 37.51 | 21.48 |
| DeQA-Score | 40.54 | 53.88 | 35.22 | 44.19 | 28.85 |
| Q-Insight | 46.05 | 54.05 | 46.65 | 52.58 | 30.90 |
| Reward/Scoring Models for Video | | | | | |
| VideoScore-v1.1 | 38.87 | 16.79 | 39.18 | **71.57** | 27.95 |
| VideoReward | 44.73 | **59.69** | 36.15 | 51.75 | 31.33 |
| UnifiedReward | 37.22 | 53.31 | 50.39 | 23.18 | 22.02 |
| VisionReward | 42.86 | 54.31 | 37.64 | 56.91 | 22.58 |
| Q-Align | 32.62 | 42.05 | 43.24 | 21.97 | 23.22 |
| AIGVE-MACS | 30.48 | 37.09 | 36.91 | 31.00 | 16.93 |
| VideoPhy2 | 29.13 | 30.75 | 24.12 | 24.00 | **37.64** |
| Dover | 42.70 | 54.27 | 44.62 | 43.69 | 28.21 |
| VideoScore2 | | | | | |
| Ours | **50.37** | 51.53 | **50.60** | 65.77 | 33.58 |

evaluation signal, we use the widely adopted video generation benchmark VBench and select a subset of dimensions, including subject consistency, background consistency, aesthetic quality, imaging quality, and motion smoothness, which are closely related to VIDEOSCORE2's visual quality and physics consistency dimensions. VBench dimensions without clear correspondence to our evaluation dimensions are excluded.

Comparison on VBench (Figure 7) shows BoN consistently outperforms random sampling, confirming that VIDEOSCORE2 provides a usable ranking signal for higher-quality selection under this automatic evaluation protocol. The Best-of-N gains are expected to be modest since all candidates in each subset are produced by the same video generative model with limited quality variance. Nevertheless, we observe consistent improvements across models and selected dimensions. We view human preference studies as a valuable direction for future validation. See full results in Appendix C.4.

### 4.6 Ablation Study

Besides ablations on SFT settings, RL training steps, as well as inference configurations (Appendix D), we conduct the following studies, providing more insights of designing VIDEOSCORE2, summarized in Table 7.

**Cold Start.** We compare RL initialized from the base Qwen2.5-VL-7B-Instruct versus the SFT checkpoint. The SFT version achieves higher average scores across both VIDEOSCORE-BENCH-v2 and OOD benchmarks, even if not superior on every benchmark. This indicates SFT provides a stronger starting point, enabling RL to focus on reward alignment rather than task formatting.

**SFT w/ and w/o rationale** We further test SFT with and without CoT-like rationales. While the CoT-based version is slightly weaker on preference benchmarks, it performs significantly better on point-score benchmarks and thus improves generalization on average. This confirms that rationales are not only important for interpretability but also beneficial for overall robustness. For preference benchmarks, we observe that CoT-style reasoning tends to produce more conservative and calibrated scores, which can

Table 7: Ablations on RL start point, rationale in SFT and score output format.

| Ablations | In-Domain | OOD Preference Benchmark | | OOD Point Score Benchmark | |
|---|---|---|---|---|---|
| | VideoScore-Bench-v2 | VideoGen-Reward-Bench | T2VQA-DB (Preference) | MJ-Bench -Video | VideoPhy2 -test |
| *I. RL start w/ and w/o SFT* | | | | | |
| RL w/ SFT (default) | **44.53** | 51.53 | 50.60 | **65.77** | **33.58** |
| RL w/o SFT | 36.70 | **54.53** | **54.54** | 56.43 | 27.69 |
| *II. SFT w/ and w/o CoT* | | | | | |
| w/ CoT (default) | **39.81** | 50.79 | 52.36 | **66.88** | **30.02** |
| w/o CoT | 32.17 | **54.74** | **58.63** | 59.06 | 21.83 |
| *III. Score output format* | | | | | |
| Normalized (default) | 44.53 | **51.53** | **50.60** | 65.77 | 33.58 |
| Raw Int Score | **45.83** | 51.19 | 30.22 | **66.51** | **34.51** |

reduce score spread and increase the tie rate under our preference. The preference benchmarks reward pairwise discriminability, whereas point-score benchmarks reward calibration, which may explain why w/o-CoT can be slightly better on OOD preference. See more discussion in Appendix D.6.

**Score format.** We ablate the output format by comparing raw integer scores and normalized float scores. While integers show slight advantages on OOD point-score benchmarks, they perform notably worse on OOD preference tasks. Using normalized float scores strikes a better balance, preserving accuracy for point-score while capturing finer quality differences in preference settings.

# 5 Conclusion

In this work, we introduced VIDEOSCORE2 for multi-dimensional, interpretable, and human-aligned evaluation of AI-generated videos. By building a comprehensive annotation pipeline that gathers diverse prompts, generative videos as well as reliable scores and rationales, we are able to train VIDEOSCORE2 in the 2-stage paradigm. Comprehensive experiments demonstrate our model outperforms existing evaluators across in-domain and out-of-domain benchmarks. We believe that VIDEOSCORE2 open a path for trustworthy evaluation and human-aligned training of generative video models. Furthermore, our evaluation also shows that model still struggle in evaluating physics and common senses in the generative models, highlighting the importance of a world model for video evaluator. We leave this as a future direction worth to explore.

# Ethics Statement

This work adheres to the ICLR Code of Ethics. In this study, no human subjects or animal experimentation was involved. All datasets used, including our curated VIDEOFEEDBACK2, were sourced and processed in compliance with relevant usage guidelines, ensuring no violation of privacy or intellectual property.

For prompt collection, we applied strict filtering to exclude NSFW, harmful, or otherwise inappropriate content, and ensured that prompts did not involve personally identifiable information (PII) or sensitive entities. Videos used for annotation were generated by publicly available text-to-video models, and only non-sensitive, safe prompts were retained. Our annotation guidelines emphasized fairness and consistency, and all annotators were trained to avoid introducing biased or discriminatory judgments.

No personally identifiable information was collected or used, and no experiments were conducted that could raise privacy or security concerns. We are committed to maintaining transparency, fairness, and integrity throughout the research process.

## Reproducibility statement

All the code and datasets used in the paper will be open-sourced after the paper is accepted. We also have provided comprehensive details for both training (see in Appendix D) and evaluation (see in Appendix C) to help the community for reproduction.

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

# Appendix

# A   Data Collection and Processing

## A.1   Collecting Text-to-Video Prompts

### Source 1: VidProM

**Rule-based filtering.**

- *NSFW probability.* The original dataset provides a probability that a given prompt may lead to NSFW content. We exclude prompts with NSFW probability greater than 0.2, following the original dataset's setting.

- *Trigger-word filtering.* Exclude prompts intended for *image-to-video* generation, which explicitly mention image attachments, and prompts specifying aspect ratios, or duration, which cannot be freely controlled in most T2V models. The trigger-word list includes: `["screen size", "16:9", "1:1", "3:4", "4k", "8k", "seconds", "message", "attach"]`.

- *Length control.* Only prompts between 15 and 100 words are retained.

**LLM Semantic filtering.**   To filter out unsuitable prompts, we use GPT-4o-mini for semantic checks and exclude problematic ones. Specifically, we remove prompts that:

- vague or meaningless, lacking a concrete task,

- containing specific people or names,

- missing substantive verbs or motion, closer to images than videos,

- describing over three actions or events, too complex for short videos.

### Source 2: Koala-36M

**Rule-based filtering.**

- Since prompts come from real video captions, we only keep those associated with video segments shorter than 5 seconds; longer captions usually describe too many actions and are unsuitable for short video generation.

- Each video–caption pair includes a *clarity score* and an *aesthetic quality score.* We exclude captions with clarity score below 0.95 or aesthetic score below 4.0.

**LLM Semantic filtering and revising**

- Same semantic checks as for VidProM, removing ambiguous or low-quality prompts.

### Source 3: OCR-Text (manually collected).

For the OCR-text category, we first drafted seed prompts that **explicitly required text to appear** in the video, then expanded them using LLMs to create realistic yet creative scenarios where text naturally integrates into the scene. These prompts are diverse and challenging, often harder to generate than purely human-written ones. For example:

- A painter adds brush strokes to a canvas, with a palette that says 'Portrait of a Lady, Acrylic Paints, Warm Tones and Fine Detail'.

- A photographer adjusts their lens with 'Capture the Perfect Shot: Photography Tips and Tricks' displayed on a screen in front of them.

In total, 200 prompts were collected.

### Source 4: Multi-Action (manually collected).

For the multi-action category, we followed a similar approach as OCR-text. We first drafted seed prompts containing **two or three connected actions**, then expanded and rewritten them with LLMs to produce diverse, story-like scenarios. In total, 200 prompts were collected, each describing a short narrative with three consecutive actions. For example:

- A woman adjusts her glasses, glances at the book with focus, and flips to the next page with a smile.

- A fluffy orange cat swats a ball of yarn, sends it rolling, then dashes after it and pounces mid-roll.

### Source 5: Camera Motion (manually collected).

For the camera-motion category, we did not generate entirely new prompts. Instead, we augmented existing prompts by appending explicit camera movement instructions at the end. Common motions include *"Zoom in," "Zoom out," "Pan left," "Pan right," "Pan up," "Pan down," "Tilt up," "Tilt down,"* and *"Tracking shot."* This simple yet effective strategy allows the dataset to capture scenarios where video realism depends on both content generation and dynamic camera behavior.

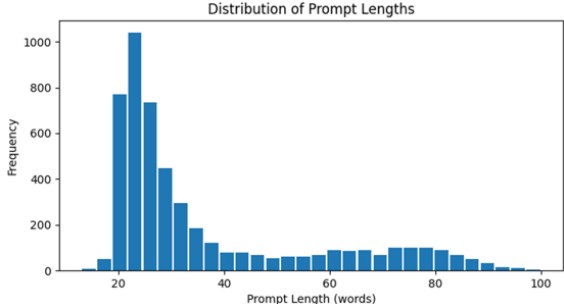

Figure 8: Distribution of Length (Num of Words) for the Prompt Set.

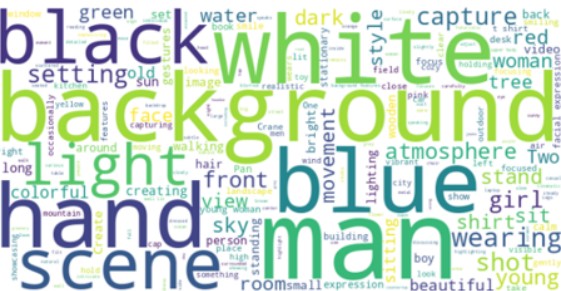

Figure 9: Word Cloud for the Prompt Set.

### A.2 Statistics of Generated Videos

We generate videos for annotation using more than twenty text-to-video (T2V) models, spanning from early diffusion-based systems such as ModelScope (Wang et al., 2023) to recent high quality generators like Kling-1.6 (Kuaishou, 2025). This ensures a broad quality spectrum, covering both weak and strong generations.

As discussed in Section 3.1, to facilitate fairer comparisons and improve annotation reliability, we categorize these models into four coarse quality tiers. For each prompt, ten videos are sampled from ten different models while maintaining a balanced distribution across the four tiers. Typically 1-2 from "Poor / Early", 3-4 from Moderate, 3-4 from "Good", and 1 from "Perfect / Modern". The resulting videos vary widely in characteristics, with durations ranging from 1 to 6 seconds, resolutions from 256×256 up to 1920×982, and frame rates from 8 to 30 fps. A full summary of the models (and its variants) used is provided in Table 8.

Table 8: Detailed information of videos in our dataset, including t2v-model sources, video fps, resolution, duration, etc.

| T2V Model (Suffix code in dataset) | Open Source | FPS | Resolution | Duration | Num | Proportion |
|---|---|---|---|---|---|---|
| **Tier1: Perfert / Modern.** 2814 videos, 10.36% | | | | | | |
| Kling-1.6 (Kuaishou, 2025) (r) | N | 24.0 | 1280*720 | 5.0s | 611 | 2.25% |
| Sora (OpenAI, 2024) (s) | N | 30.0 | 1920*982 | 10.0s | 298 | 1.10% |
| Pika-2.2 (Pika-Labs, 2025) (t) | N | 24.0 | 1280*720 | 5.0s | 321 | 1.18% |
| StepVideo-T2V (Ma et al., 2025a) (y) | Y | 25.0 | 992*544 | 4.0s | 741 | 2.73% |
| Wanx-2.1 (14B) (Wan et al., 2025) (w) | Y | 25.0 | 832*480 | 3.9s | 281 | 1.03% |
| Ruyi (Team, 2024a) (A) | Y | 24.0 | 1008*576 | 5.0s | 184 | 0.68% |
| CogVideoX-1.5 (Yang et al., 2024) (g) | Y | 10.0 | 1360*768 | 4.0s | 378 | 1.39% |
| **Tier2: Good.** 9598 videos, 33.53% | | | | | | |
| Wanx-2.1 (1.3B) (Wan et al., 2025) (v) | Y | 24.0 | 832*480 | 3.9s | 1497 | 5.51% |
| MagicTime (Yuan et al., 2025) (q) | Y | 8.0 | 512*512 | 2.0s | 1741 | 6.41% |
| Mochi1-Preview (Team, 2024b) (c) | Y | 10.0 | 848*480 | 1.9s | 1649 | 6.07% |
| LaVie-base (Wang et al., 2024a) (h) | Y | 8.0 | 512*320 | 2.0s | 1547 | 5.69% |
| CogVideoX (5B) (Yang et al., 2024) (f) | Y | 10.0 | 720*480 | 4.0s | 1786 | 6.57% |
| OpenSora-Plan (v1.3) (Lin et al., 2024a) (u) | Y | 18.0 | 640*352 | 5.2s | 1378 | 5.07% |
| **Tier3: Moderate.** 11349 videos, 41.77% | | | | | | |
| CogVideoX (2B) (Yang et al., 2024) (e) | Y | 10.0 | 720*480 | 4.0s | 1774 | 6.53% |
| LTX-Video-0.9.5 (HaCohen et al., 2024) (z) | Y | 25.0 | 704*480 | 4.8s | 1692 | 6.23% |
| OpenSora (v1.2) (Zheng et al., 2024b) (x) | Y | 8.0 | 640*480 | 1.6s | 907 | 3.34% |
| Latte (Ma et al., 2025c) (b) | Y | 10.0 | 512*512 | 1.6s | 1510 | 5.56% |
| VideoCrafter2 (Chen et al., 2024) (n) | Y | 10.0 | 512*320 | 1.6s | 1172 | 4.31% |
| Vchitect-2.0 (Fan et al., 2025) (p) | Y | 10.0 | 512*320 | 1.6s | 1235 | 4.55% |
| AnimateDiff (Guo et al., 2024) (a) | Y | 10.0 | 512*512 | 2.4s | 1755 | 6.46% |
| Hotshot-XL (Mullan et al., 2023) (m) | Y | 8.0 | 673*384 | 1.0s | 1304 | 4.80% |
| **Tier4: Poor / Early.** 3407 videos, 12.54% | | | | | | |
| ModelScope (Wang et al., 2023) (d) | Y | 10.0 | 256*256 | 2.4s | 967 | 3.56% |
| LTX-Video-0.9.1 (HaCohen et al., 2024) (i) | Y | 10.0 | 704*512 | 3.3s | 1333 | 4.91% |
| ZeroScope (Sterling, 2024) (j) | Y | 10.0 | 256*256 | 2.4s | 395 | 1.45% |
| T2V-Zero (Khachatryan et al., 2023) (k) | Y | 10.0 | 256*256 | 0.8s | 712 | 2.62% |
| **All: 27168 videos** | | | | | | |

### A.3 Video Examples from Different Quality Tiers.

Below we show some videos of each quality tier, from "Perfect / Modern" to "Poor / Early".

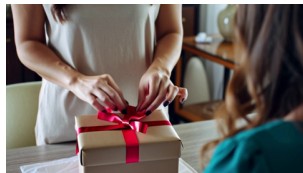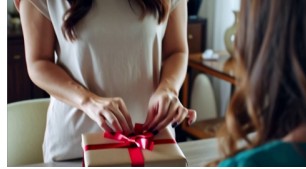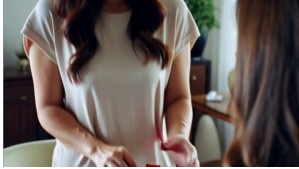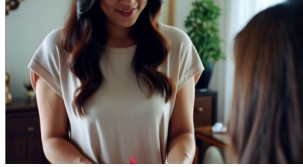

Example video of the quality tier **"Perfect / Modern"**, from Kling-1.6. Prompt is: *A woman ties a red ribbon around a gift box, carefully wraps it in shiny paper, and then smiles as she hands it to her friend.*

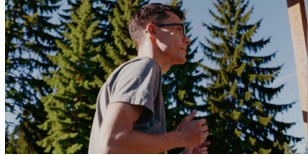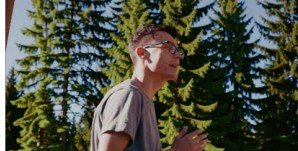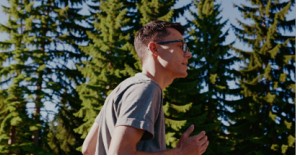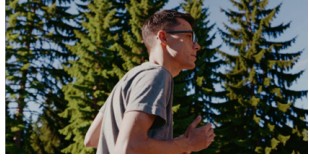

Example video of the quality tier **"Perfect / Modern"**, from Sora. Prompt is: *A young man in glasses and a gray t-shirt stands on a wooden deck, gesturing with his hands and expressing different emotions. The background shows a scenic forest with tall trees and a clear sky. His facial expressions change as he moves his hands, sometimes near his face, indicating various reactions. The camera captures him in a steady medium shot, focusing on his upper body and gestures.*

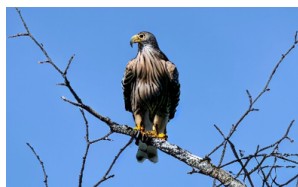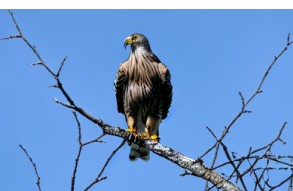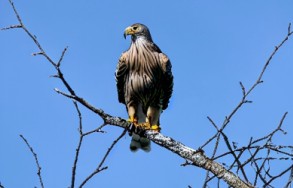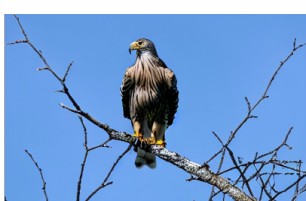

Example video of the quality tier **"Good"**, from LaVie-base. Prompt is: *A hawk perches on a leafless tree branch, facing away from the camera and gazing up at the clear blue sky. The calm scene features a few wispy clouds and barren tree branches. The hawk remains still, with occasional head movements, set against a peaceful, natural backdrop.*

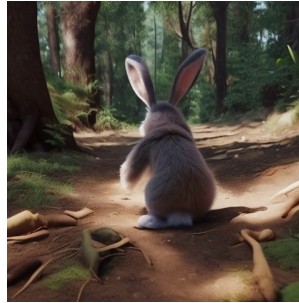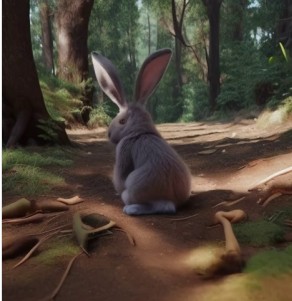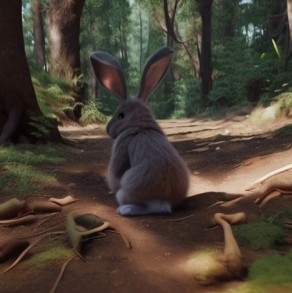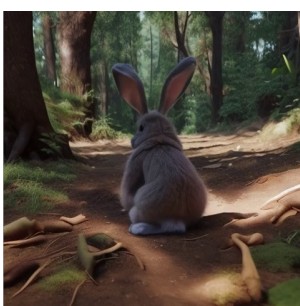

Example video of the quality tier **"Good"**, from MagicTime. Prompt is: *A scene showing the lost bunny, its eyes wide with fear, as it navigates through a dense forest, with Sammy guiding it safely home.*

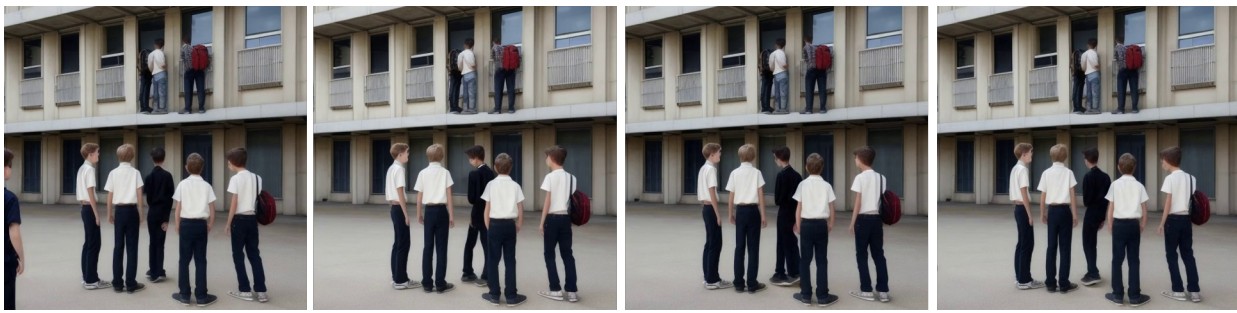

Example video of the quality tier **"Moderate"**, from AnimateDiff. Prompt is: *5 boys of age 17 standing outside a school building. Three of them are looking at other students passing by, one is looking at his mobile, and two are talking to each other. Crane up.*

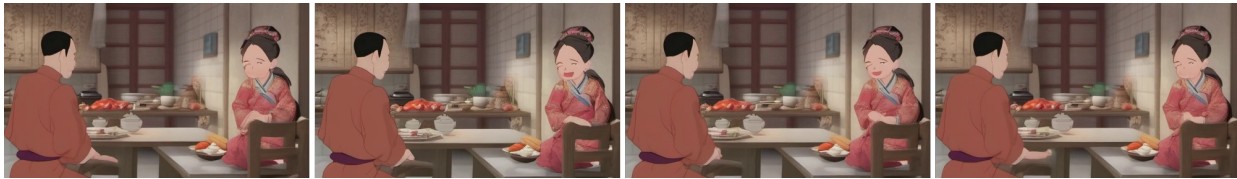

Example video of the quality tier **"Moderate"**, from Hotshot-XL. Prompt is: *A cozy family kitchen with breakfast items on the table. Xiao Ming, wearing traditional home attire, engages in a lively conversation with his parents, emphasizing the warmth of family bonds and traditional values.*

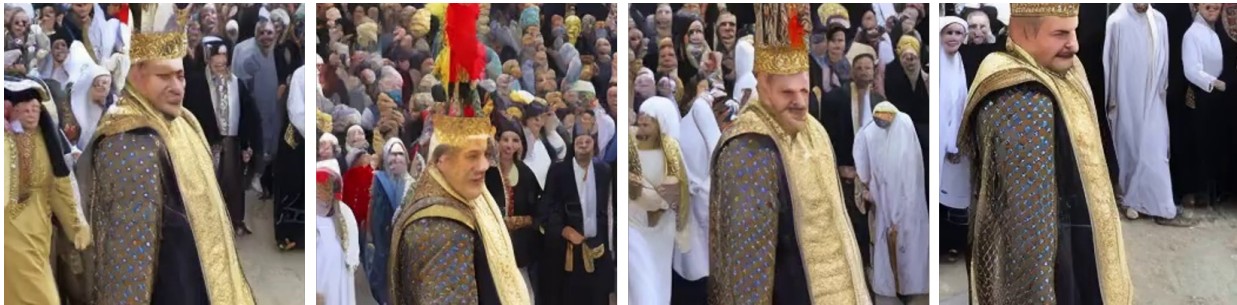

Example video of the quality tier **"Poor / Early"**, from ModelScope. Prompt is: *The Kurdish king wears a crown of gold on his head in 1850. He is imposing, serious, authoritative, loving, tall, and handsome. He walks among the people in Kurdish clothes. Tilt down.*

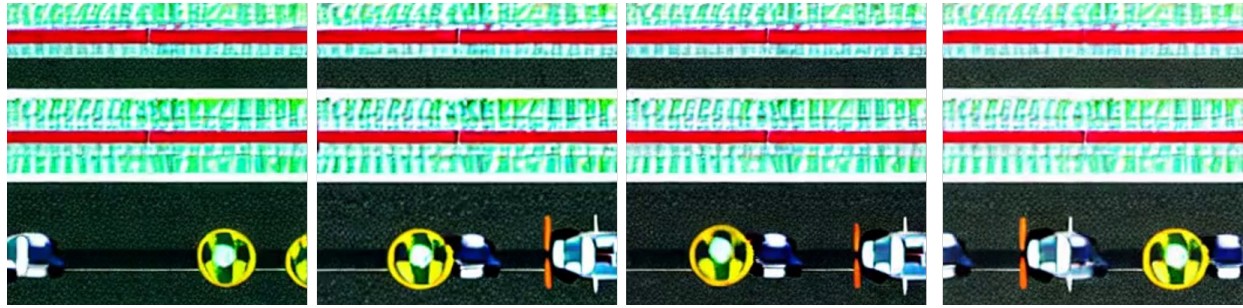

Example video of the quality tier **"Poor / Early"**, from Text2Video-Zero. Prompt is: *A busy highway with cars and trucks moving in both directions under a clear blue sky. The scene, filmed from a moving vehicle, highlights a white van with 'Martinez returns from Florida' on its side.*

### A.4 Annotation Details.

**Main Instructions**   The main instruction required annotators to assign a score for each dimension based on its definition and to provide a short comment describing the issues observed. For instance, under *Visual Quality*, comments could include "low resolution," "local blur," or "brightness flicker." For *Text-to-Video Alignment*, annotators were asked to note missing elements from the prompt, while for *Physical/Common-Sense Consistency*, they were instructed to highlight any violations of physical laws, common sense, or abnormal artifacts.

**Detailed Guidelines**   In addition, we provided detailed annotation guidelines to ensure consistency: (i) if a dimension was rated 5, the comment could be omitted, since in such cases a template-based rationale could be generated; (ii) if a video was entirely black or unrecognizable, it should be skipped. Dimension-specific clarifications were also given:

- **Visual Quality:** Videos scoring 5 should look almost perfect, comparable to real footage; while videos scoring 1 corresponds to severe flaws, where the subject, object, or motion is hardly identifiable, or strong distortion/disconnection is present.

- **Text-to-Video Alignment:** For prompts with multiple actions (e.g., "Open the refrigerator, put the elephant in, and close the door"), all actions must be checked for faithful realization. While alignment often correlates with visual quality, clear and smooth videos may still fail to match the prompt. Annotators were instructed to focus on whether the prompt content was expressed correctly, ignoring minor extra details unless they severely misled the meaning.

- **Physical/Common-Sense Consistency:** Most videos contain at least minor physical issues, but the severity varies. If a prompt itself is unrealistic or absurd, annotators were instructed to disregard this and judge the video independently. Complex reasoning was unnecessary; everyday common sense was considered sufficient for evaluation.

Furthermore, annotators are informed that **each batch of 10 videos they see sequentially corresponded to the same prompt** but came from different T2V models with diverse quality levels, enabling fairer and more calibrated scoring.

**Human Inspection on Rationale Quality**   We conducted a human evaluation on 200 sampled rationales, assessing faithfulness to the original human comments and correctness against video content:

| Aspect | Good | Moderate | Bad |
|---|---|---|---|
| **Faithfulness** | 74.50% | 23.00% | 2.50% |
| **Correctness** | 81.50% | 16.50% | 2.00% |

Table 9: Human inspection results for rationale quality on sampled entries.

The Bad rate is low (2.5% for faithfulness, 2% for correctness), indicating that hallucination is not a prevalent issue. The Moderate cases in faithfulness are primarily partial omissions of details from human comments rather than fabricated content.

### A.5 Prompt Templates for Annotation Processing

Table 10 shows the prompt template in LLM augmented scoring for eliciting detailed thinking from human annotated quality comments. Table 11 shows the prompt template for revising analysis process when the human-annotated score and then adjusted model score are inconsistent with the thinking model's output analysis. Table 12 shows the prompt template for building query in SFT data and running inference.

Table 10: Prompt template in LLM augmented scoring for eliciting detailed thinking from human annotated quality comments.

---

We are collecting and processing human annotations for the quality evaluation of AI-generated videos.

**Dimension definitions:**

(1) Visual Quality:

Mainly evaluates the video's visual and optical properties, including 'resolution, overall clarity, local blurriness, smoothness, stability of brightness/contrast, distortion/misalignment, abrupt changes, and any other factors the affect the watching experience'. The keywords written by the annotators are also mostly derived from the above factors.

(2) Text Alignment:

Mainly assesses whether the generated video fully and accurately depicts the elements mentioned in the text prompt, such as characters, actions, animals, etc., as well as background, quantity, color, weather, and so on. So the keywords written by annotators sometimes only indicate the elements that are missing from the video.

(3) Physical/Common-sense Consistency:

Mainly examines whether there are any violations of common sense, physical laws, or any other aspects in the video that appear strange or unnatural. Most of the keywords provided by annotators point out the specific abnormalities or inconsistencies they observed in the video.

With the reference of some frames of the video, and the comments of 3 dimensions from a human annotator may also be provided, please do your best to analyze and give a INTEGAR score between 1 and 5 for these dimensions, where 1 means very bad, 3 means medium, and 5 means very good.

Sometimes human comments may be brief or lacking details, or the human comments may be null, — please check the aspects in dimension definitions and make sure to thoroughly perceive and analyze the video on your own. **Your reasoning should be detailed, professional, and comprehensive. \*\*DO NOT mention any human comment in your thinking\*\***; you should pretend not to know these comments (if they are provided), they are provided solely to inform and enhance your understanding for better evaluation.

**Output format**:

Your response must follow the format below strictly:

{

"score_visual": "quality score" (this field is only allowed to be a number between 1 and 5, inclusive, ),

"score_t2v": "quality score" (this field is only allowed to be a number between 1 and 5, inclusive),

"score_phy": "quality score" (this field is only allowed to be a number between 1 and 5, inclusive),

}

DO NOT include any text before or after the json block.

**Here is the Input:**

Text prompt used to generate the video: $prompt

Comment for "visual quality": $comment_visual

Comment for "text-to-video alignment" (the elements or events not expressed or not aligned in the video): $comment_t2v

Comment for "physical/common-sense consistency" (the elements or events that look weird, abnormal or unnatural): $comment_phy

---

Table 11: Prompt template for revising analysis process when the human-annotated score and then adjusted model score are inconsistent with the thinking model's output analysis..

---

I'm conducting a multi-dimensional quality assessment of AI-generated videos, focusing on the dimensions of (1) Visual Quality, (2) Text Alignment, and (3) Physical/Common-sense Consistency.

I will provide a multi-dimensional quality analysis for a video. However, the scores assigned in the analysis may not be entirely accurate. And the ground truth scores for each dimension will also be provided. Your task is to adjust the analysis text accordingly to ensure it aligns with the actual scores. In many cases, this means revising the severity of issues for certain dimension based on the ground truth scores. The scale of score is [1, 2, 3, 4, 5].

**Important Notes:**

(1) **Any human comment should NOT be mentioned in the output analysis**. If the input analysis quote or mention human comments, you should pretend not to know them in your output, they are provided solely to inform and enhance your understanding for better evaluation.

(2) **DO NOT** alter the overall structure or core meaning of the analysis. Only revise specific expressions or phrases as needed so that the content reasonably reflects the provided scores.

(3) The input original analysis is constructed from the sampled frames of the video, if the input analysis includes evaluations of individual frames or frame-by-frame assessments, you should appropriately transform them into an overall evaluation of the entire video, since the final output is expected to be based on the video as a whole.

(4) Your output analysis should be approximately the same length as the input analysis. If the input analysis is not very detailed and specific, you may extend your output accordingly.

**Output format:**

Your response must follow the format below strictly:

{ "new_thinking": "modified analysis" (this field is only allowed to be string), }

DO NOT include any text before or after the dictionary block.

**Here is the input:**

multi-dimensional analysis: $thinking
ground truth score of Dim-1 "Visual Quality":$v_score

ground-truth scoreof Dim-2 "Text-to-Video Alignment":$t_score

ground-truth of Dim-3 "Physical Consistency' (also referred to as Common-sense Consistency): $p_score

---

Table 12: Prompt template for building query in SFT data and running inference.

---

We would like to evaluate its quality from three dimensions: 'visual quality', 'text-to-video alignment' and 'physical/common-sense consistency'.

Below is the definition of each dimension:

(1) visual quality:

The dimension 'visual quality' cares about the video's visual and optical properties, including 'resolution, overall clarity, local blurriness, smoothness, stability of brightness/contrast, distortion/misalignment, abrupt changes, and any other factors the affect the watching experience'. The keywords written by the annotators are also mostly derived from the above factors.

(2) text alignment:

The dimension 'text-to-video alignment' mainly assesses whether the generated video fully and accurately depicts the elements mentioned in the text prompt, such as characters, actions, animals, etc., as well as background, quantity, color, weather, and so on. So the keywords written by annotators sometimes only indicate the elements that are missing from the video.

(3) physical/common-sense consistency:

The dimension 'physical/common-sense consistency' mainly examines whether there are any violations of common sense, physical laws, or any other aspects in the video that appear strange or unnatural. Most of the keywords provided by annotators point out the specific abnormalities or inconsistencies they observed in the video.

Here we provide an AI video generated by text-to-video models and its text prompt:

$t2v_prompt.

Based on the video content and the dimension definitions, please evaluate the video quality and give the quality score. The score must be in the range of 1 - 5.

---

# B  Evaluation Suite

## B.1  Dimension Matching and Modification in Out-of-Domain Benchmarks

Since different benchmarks define varying dimensions and scoring scales, we align them with the three evaluation dimensions of VIDEOSCORE2 (visual quality, text alignment, and physical consistency) and, where necessary, rescale their ground-truth scores. Below we summarize the mapping rules for each benchmark.

**VideoGenReward Bench.**  This is a pairwise preference benchmark containing 4,691 videos, forming 25,234 video pairs. It evaluates three dimensions—visual quality (VQ), text alignment (TA), and motion quality (MQ)—and also provides an *Overall* preference label indicating which video is better overall. Among these, VQ and TA correspond closely to VIDEOSCORE2 's first two dimensions (despite slight definitional differences), so these two dimensions are used for this benchmark. For the *Overall* preference, we use the mean of all available dimension scores from VIDEOSCORE2 or the baseline method (if the baseline only has one quality score output, then that score is used directly).

**T2VQA-DB.**  Originally a human-annotated video quality dataset with 10,000 videos, each labeled with a single quality score in the range [1,100]. We sample 2,000 videos and construct 1,822 pairs by comparing human-annotated scores. Since the dataset provides only one dimension (the final score), we predict preference by averaging all dimension scores from VIDEOSCORE2 or the baseline method (if the baseline only has one quality score output, then that score is used directly).

**MJ-Bench-Video.**  This benchmark contains 2,170 videos and adopts a point-score format with five dimensions: *fineness*, *alignment*, *consistency & coherence*, *safety*, and *bias & fairness*. We select the first three, which correspond to VIDEOSCORE2 's three evaluation dimensions. For baselines with only one final score, we "broadcast" this score across multiple dimensions. The benchmark uses a {0,1,2} scale, whereas VIDEOSCORE2 and other baselines output (or are normalized to) integer scores in [1,5]. Thus, we apply the following mapping, where $x$ denotes the original score of each dimension, $v$, $t$, $p$ denote the rescaled score of for "visual quality", "text alignment" and "physical consistency", respectively:

$$v = \begin{cases} 0 & \text{if } x \in \{1,2\}, \\ 1 & \text{if } x \in \{3,4\}, \\ 2 & \text{if } x = 5, \end{cases} \quad t = \begin{cases} 0 & \text{if } x = 1, \\ 1 & \text{if } x \in \{2,3\}, \\ 2 & \text{if } x \in \{4,5\}, \end{cases} \quad p = \begin{cases} 0 & \text{if } x = 1, \\ 1 & \text{if } x \in \{2,3\}, \\ 2 & \text{if } x \in \{4,5\}. \end{cases}$$

The benchmark also provides an *Overall* score, for which we again take the mean of all available dimension scores (or the single dimension if only one is provided), rescaled into {0,1,2} using the same rule.

**VideoPhy2-test.**  This benchmark contains 3,396 videos with two dimensions, SA: *semantic adherence* and PC: *physical consistency*. These map perfectly to VIDEOSCORE2 's second and third dimensions. For baselines lacking one of the dimensions (e.g., VideoReward, which provides VQ, TA, MQ but no physical consistency), we skip the missing dimension. The scoring scale is {1,2,3,4,5}, so no rescaling is required.

**B.2 Dimension Matching and Score Rescaling for Baselines**

Since different baseline models adopt varying evaluation dimensions and scoring scales, we apply **dimension matching** and **score rescaling** to make them compatible with VIDEOSCORE-BENCH-V2 and VIDEOSCORE2. Our goal is to ensure that all baselines output scores on the three dimensions—visual quality (v), text alignment (t), and physical consistency (p)—within a unified range of integers 1-5. A summary of the mapping rules is provided in Table 13.

**Baseline Dimension Matching.** With $v$, $t$, and $p$ to denote the score of visual quality, text alignment, physical/common-sense consistency, respectively, we consider three cases:

- **Broadcast.** Some baselines only output a single final score. In this case, we broadcast the same score to our three dimensions $v$, $t$, and $p$.

- **Good Match.** Some baselines already report dimensions that closely match ours, so we directly use their outputs without modification.

- **Customized.** For baselines with different or partially overlapping dimensions, we design customized mappings.
  - *VideoReward*: outputs Visual Quality, Text Alignment, and Motion Quality. We use the outputs of first two dimensions as $v$ and $t$, and skip Motion Quality.
  - *AIGVE-MACS*: outputs multiple fine-grained dimensions. We average {technical quality, element quality, action quality} as $v$, average {element presence, action presence} as $t$, and use physics as $p$.
  - *VideoPhy2-Auto-Eval*: outputs Semantic Adherence (SA) and Physical Consistency (PC). We use SA as $t$ and PC as $p$, while skipping $v$.

**Baseline Score Rescaling.** To make results comparable, we rescale all baseline outputs into a unified integer range of 1–5. A summary of the mapping rules is provided in Table 13.

- Linear Scaling or No Scaling. For baselines with well-defined score ranges (e.g., [0,1], [0,100]), we apply linear normalization followed by rounding to the nearest integer in $\{1, 2, 3, 4, 5\}$.

- Ordinal categories using Gaussian-distribution quantile thresholds. For baselines without fixed score bounds, we adopt an ordinal mapping based on Gaussian-distribution quantile thresholds. Specifically, raw scores are assumed to approximately follow a Gaussian distribution and are divided into five categories using the 20%, 40%, 60%, and 80% quantiles of the standard normal distribution. If the raw scores typically fall within [-2.0, 2.0] and we assume a Gaussian Distribution $N(0,1)$, thus apply the following mapping:

$$
\text{score} = \begin{cases} 1 & \text{if } z < \Phi^{-1}(0.2), \\ 2 & \text{if } \Phi^{-1}(0.2) \leq z < \Phi^{-1}(0.4), \\ 3 & \text{if } \Phi^{-1}(0.4) \leq z < \Phi^{-1}(0.6), \\ 4 & \text{if } \Phi^{-1}(0.6) \leq z < \Phi^{-1}(0.8), \\ 5 & \text{otherwise}, \end{cases}
$$

where $z$ is the raw model score and $\Phi^{-1}$ denotes the inverse CDF of the standard Gaussian.

  - *ImageReward* and *VisionReward*: most scores are in [-2.0, 2.0], assume $N(0,1)$ and follow the mapping above.
  - *VideoReward*: most scores are in [-3.0, 3.0], so we assume a Gaussian Distribution $N(0,1.5)$, and $z$ is replaced by $z/1.5$ in the rules above to firstly normalize the raw score before converting it to integers.

Table 13: Rescale output scores and map dimensions of baselines models to align with our VIDEOSCORE2 and VIDEOSCORE-BENCH-V2.

| Model | Dimension Mapping | Original Scale | Score Rescaling Method |
|---|---|---|---|
| **Reward/Scoring Models for Image** (averaged on sampled frames) | | | |
| ImageReward | Broadcast | most in [-2.0,2.0] | Ordinal categories using Gaussian-distribution quantile thresholds. |
| DeQA-Score | Broadcast | [0.0, 5.0] | Linearly amplify and round |
| Q-Insight | Good Match | [1.0, 5.0] | Linearly amplify and round |
| **Reward/Scoring Models for Video** | | | |
| VideoReward | Customized | most in [-4.0,4.0] | Ordinal categories using Gaussian-distribution quantile thresholds. |
| UnifiedReward | Good Match | {1,2,3,4,5} | No rescaling |
| VisionReward | Broadcast | most in [-1.0,1.0] | Ordinal categories using Gaussian-distribution quantile thresholds. |
| Q-Align | Broadcast | [0.0, 1.0] | Linearly amplify and round |
| AIGVE-MACS | Customized | {1,2,3,4,5} | No rescaling |
| VideoPhy2 | Customized | {1,2,3,4,5} | No rescaling |
| Dover | Broadcast | [0.0, 1.0] | Linearly amplify and round |

## C   Full Evaluation Results

Besides the in-domain benchmark VIDEOSCORE-BENCH-V2, We evaluate VIDEOSCORE2 on four Out-of-Domain (OOD) benchmarks: VideoGen-Reward-Bench, T2VQA-DB-Preference, MJ-Bench-Video and Video-Phy2-test, the first two are pairwise preference benchmarks while the last two supports point-score.

Among the four OOD benchmarks, T2VQA-DB-Preference is single-dimensional (only final quality score), while the other three have multiple dimensions. So we include the detailed full results of these three benchmarks in the following pages.

### C.1   Full Results on VideoGen-Reward-Bench

VideoGen-Reward-Bench is a video preferenc over three dimensions: visual quality, text alignment, and motion quality. The task is to compare a pair of videos and judge which one is better along these axes. Among them, the first two dimensions are broadly aligned with ours, while the benchmark also provides an additional measure of overall preference.

For the preference benchmarks, we report results under two settings. The w/ ties version includes all test entries, where in some cases the two compared videos (including the ground-truth reference) are judged as equally preferred. The w/o ties version is a subset obtained by removing those entries with equal preference labels. The full evaluation results of preference prediction accuracy are shown in Table 14.

Table 14: Full evaluation results on **VideoGen-Reward-Bench**. **Bold** denotes the best model and the underlined denotes the second best.

| VideoGen-Reward-Bench | Visual Quality | | Text Alignment | | Overall | |
|---|---|---|---|---|---|---|
| | w ties | w/o ties | w ties | w/o ties | w ties | w/o ties |
| **Reward/Scoring Models for Image** (averaged on sampled frames) | | | | | | |
| ImageReward | 31.64 | 51.40 | 44.00 | 60.72 | 47.14 | 58.61 |
| DeQA-Score | 41.07 | 69.55 | 36.22 | 53.23 | 53.88 | 67.91 |
| Q-Insight | 30.68 | 66.34 | 42.11 | 59.47 | 54.05 | 66.34 |
| **Reward/Scoring Models for Video** | | | | | | |
| VideoScore-v1.1 | 47.41 | 30.84 | 26.09 | 30.85 | 16.79 | 40.19 |
| VideoReward | **53.21** | **75.58** | **52.75** | **72.18** | **59.69** | **73.66** |
| UnifiedReward | 41.27 | 39.42 | 40.11 | 36.58 | 53.31 | 58.83 |
| VisionReward | 35.89 | 59.03 | 44.86 | 61.15 | 54.31 | 67.58 |
| Q-Align | 32.01 | 52.98 | 35.77 | 51.06 | 42.05 | 52.52 |
| AIGVE-MACS | 38.05 | 30.80 | 30.76 | 11.66 | 37.09 | 37.08 |
| VideoPhy2 | - | - | 37.04 | 22.14 | 30.75 | 26.41 |
| Dover | 39.34 | 68.87 | 38.01 | 55.65 | 54.27 | 68.58 |
| **Ours** | | | | | | |
| VIDEOSCORE2 (SFT only) | 37.74 | 63.17 | 43.07 | 61.35 | 50.79 | 63.80 |
| VIDEOSCORE2 (RL w/o SFT) | 34.67 | 65.87 | 48.70 | 65.92 | 54.53 | 65.59 |
| VIDEOSCORE2 (SFT + RL) | 37.44 | 63.08 | 42.87 | 60.61 | 51.53 | 63.72 |

## C.2 Full Results on MJ-Bench-Video

To maximize compatibility with the evaluation dimensions of VIDEOSCORE2, we selected three aspects from MJ-Bench-Video that are most semantically aligned: Fineness, Alignment, and Coherence & Consistency. These aspects correspond respectively to the three dimensions in VIDEOSCORE2: visual quality, text alignment, and physical/commonsense consistency.

The full evaluation results of the three aspects and the overall scores are shown in Table 15, with prediction accuracy between model outputs and ground truths adopted as metrics.

Table 15: Full evaluation results on **MJ-Bench-Video**. **Bold** denotes the best model and the underlined denotes the second best.

| MJ-Bench-Video | Accuracy | | | |
| --- | --- | --- | --- | --- |
| | Fineness | Alignment | Coherence & Consistency | Overall |
| **Reward/Scoring Models for Image** (averaged on sampled frames) | | | | |
| ImageReward | 47.05 | 28.07 | 29.03 | 37.51 |
| DeQA-Score | 18.57 | 51.20 | 52.40 | 44.19 |
| Q-Insight | 12.72 | 42.86 | 28.07 | 52.58 |
| **Reward/Scoring Models for Video** | | | | |
| VideoScore-v1.1 | 13.69 | **64.19** | **79.22** | **71.57** |
| VideoReward | **79.36** | 38.99 | - | 51.75 |
| UnifiedReward | 43.50 | 21.98 | 18.16 | 23.18 |
| VisionReward | 36.31 | 55.99 | 67.51 | 56.91 |
| Q-Align | 14.77 | 31.74 | 26.41 | 21.97 |
| AIGVE-MACS | 20.18 | 26.27 | 21.39 | 31.00 |
| VideoPhy2-Auto-Eval | - | 38.97 | 7.89 | 24.00 |
| Dover | 29.26 | 45.67 | 48.02 | 43.69 |
| **Ours** | | | | |
| VIDEOSCORE2 (SFT only) | 33.95 | 46.20 | 57.80 | 66.88 |
| VIDEOSCORE2 (RL w/o SFT) | 64.68 | 32.79 | 57.27 | 56.43 |
| VIDEOSCORE2 (SFT + RL) | 22.50 | 48.58 | 66.79 | 65.77 |

### C.3 Full Results on VideoPhy2-test

Video-Phy2-Test is a human-annotated test set with two dimensions: semantic adherence and physical consistency (abbreviated as semantic and physical in our tables). These two dimensions correspond directly to the latter two evaluation dimensions in our framework.

The full evaluation results of the two dimensions are shown in Table 16, with prediction accuracy and PLCC between model outputs and ground truths adopted as metrics.

Table 16: Full evaluation results on **Video-Phy2-test**. **Bold** denotes the best model and the underlined denotes the second best.

| Video-Phy2-test | Accuracy | | | PLCC | | |
|---|---|---|---|---|---|---|
| | Semantic | Physical | Avg | Semantic | Physical | Avg |
| **Reward/Scoring Models for Image** (averaged on sampled frames) | | | | | | |
| ImageReward | 23.73 | 19.23 | 21.48 | 15.28 | 3.07 | 9.18 |
| DeQA-Score | 28.74 | 28.96 | 28.85 | 3.55 | 2.14 | 2.85 |
| Q-Insight | 29.21 | 32.59 | 30.90 | 22.45 | 4.98 | 13.72 |
| **Reward/Scoring Models for Video** | | | | | | |
| VideoScore-v1.1 | 29.81 | 26.08 | 27.95 | 11.61 | 13.09 | 12.35 |
| VideoReward | 31.33 | - | 31.33 | 34.54 | - | **34.54** |
| UnifiedReward | 17.64 | 26.39 | 22.02 | 34.57 | 22.78 | 28.68 |
| VisionReward | 31.95 | 13.20 | 22.58 | 28.11 | 13.67 | 20.89 |
| Q-Align | 18.43 | 28.00 | 23.22 | 5.52 | 2.70 | 4.11 |
| AIGVE-MACS | 12.23 | 21.63 | 16.93 | 8.09 | 11.90 | 10.00 |
| VideoPhy2-Auto-Eval | **37.96** | **37.31** | **37.64** | 38.64 | **29.84** | 34.24 |
| Dover | 26.56 | 29.86 | 28.21 | 3.85 | 1.15 | 2.50 |
| **Ours** | | | | | | |
| VIDEOSCORE2 (SFT only) | 32.24 | 27.80 | 30.02 | 27.22 | 13.85 | 20.54 |
| VIDEOSCORE2 (RL only) | 31.71 | 23.66 | 27.69 | 39.07 | 16.90 | 27.99 |
| VIDEOSCORE2 (SFT + RL) | 37.48 | 29.67 | 33.58 | **41.08** | 17.57 | 29.33 |

## C.4  Full Results of Best-of-N sampling on VBench

Table 17: Quality evaluatioin of eight T2V models on V-Bench **with** BoN sampling by our VIDEOSCORE2, compared with random ones. We can see consistent improvement.

| Best-of-N | Dimensions in **VBench** | | | | | | | | | | | |
|---|---|---|---|---|---|---|---|---|---|---|---|---|
| | **Average** | | Subject | | Background | | Aesthetic | | Imaging | | Motion | |
| | Random | **BoN** | Random | **BoN** | Random | **BoN** | Random | **BoN** | Random | **BoN** | Random | **BoN** |
| Lavie-base | 82.85 | **83.07** | 95.40 | **95.69** | 96.89 | **97.08** | 56.64 | **57.15** | **67.99** | 67.98 | 97.34 | **97.47** |
| AnimateDiff | 81.97 | **83.15** | 91.16 | **94.18** | 94.30 | **95.64** | **60.90** | 60.28 | **69.36** | 69.01 | 94.14 | **96.64** |
| VideoCrafter1 | 80.03 | **80.63** | 95.35 | **95.58** | 95.76 | **96.05** | 46.00 | **47.67** | 67.03 | **67.58** | 95.99 | **96.26** |
| ModelScope | 78.75 | **79.70** | 93.68 | **95.07** | 95.52 | **96.40** | 46.23 | **47.60** | 61.64 | **62.32** | 96.66 | **97.10** |
| ZeroScope | 76.36 | **77.84** | 91.32 | **93.04** | 94.50 | **95.37** | 45.27 | **47.55** | 55.25 | **56.95** | 95.48 | **96.30** |
| LVDM | 75.33 | **76.26** | 88.79 | **89.91** | 93.14 | **93.81** | 41.01 | **42.00** | 60.94 | **62.24** | 92.75 | **93.35** |

## C.5  Inference Cost

| Method | Inference Time (s) | Tokens (Input + Output, k) |
|---|---|---|
| **VideoScore2** | 24.1 | 1605 + 418 |
| **VideoReward** | 1.6 | 1610 + 0 |
| **VisionReward** | 10.3 | 8411 + 58 |
| **UnifiedReward** | 2.7 | 2447 + 34 |
| **VideoPhy2-AutoEval** | 4.1 | 355 + 8 |
| **ImageReward** | 41.3 | 3792 + 2 |

Table 18: Inference time and token usage across our method and baselines. For ImageReward, the sampling rate is 2fps, with max 16 frames. Other baselines adopt the default configuration.

# D  Experiment Setup and Ablation Studies

## D.1  SFT experiment setup

We conduct SFT with sampling fps of 2, a maximum frame resolution of 960×720, learning rates of 5e-5, and epochs of 2 with one epoch taking about 6 hours on 8×A800 GPUs.

## D.2  Ablation on sampling fps in SFT training

During training, videos are sampled at 2 fps, which we find sufficient for evaluation: "visual quality" primarily reflects global perceptual properties, "text alignment" focuses on semantic adherence, and most issues of "physical consistency" or abnormal events typically last longer than half a second, ensuring they can still be captured at this frame rate.

We also conduct an ablation on a 17k subset to study the effect of training sampling fps, comparing 2, 4, and 8 fps settings. As shown in Table 19, increasing the sampling rate does not yield significant performance gains, while it noticeably increases computational cost and training time. Therefore, we adopt 2 fps as the default setting in our main SFT experiments and in all subsequent ablations of other hyper-parameters.

Table 19: Ablation results on a 17k subset of VIDEOSCORE2 data for different sampling fps in SFT.

| Train Sampling fps | Accuracy | | | | Relaxed Accuracy | | | | PLCC | | | |
|---|---|---|---|---|---|---|---|---|---|---|---|---|
| (17k subset) | Visual | Align | Phy | Avg | Visual | Align | Phy | Avg | Visual | Align | Phy | Avg |
| 2fps | 54.67 | 39.33 | 46.67 | 46.89 | 94.00 | 81.33 | 90.00 | 88.44 | 73.62 | 60.24 | 54.72 | 62.86 |
| 4fps | 48.67 | 42.67 | 49.33 | 46.89 | 94.00 | 82.00 | 92.67 | 89.56 | 67.87 | 61.85 | 63.86 | 64.53 |
| 8fps | 51.00 | 45.33 | 48.00 | 48.11 | 92.00 | 85.33 | 88.67 | 88.67 | 64.34 | 65.71 | 52.05 | 60.70 |

### D.3 Ablation on learning rate and epochs in SFT training

We perform ablations on two key hyper-parameters: learning rate 1e-5, 2e-5, 5e-5, 1e-4, 2e-4 and epochs $\{1, 2, 3\}$. The results on VIDEOSCORE-BENCH-V2 are summarized in Table 20.

For learning rate, $1 \times 10^{-4}$ achieves slightly higher accuracy than $5 \times 10^{-5}$, but its loss curve is less stable and shows lower values in the second epoch, as shown in Figure 10 and 11, suggesting potential overfitting, which could harm performance on out-of-domain benchmarks. By contrast, 2e-5 exhibits a much higher loss curve in later stages, indicating underfitting. Balancing in-domain accuracy and loss smoothness, we choose 5e-5 as the default learning rate.

For epochs, the 2-epoch setting outperforms both 1 and 3 epochs, and is therefore adopted as the main version. This chosen SFT checkpoint also serves as the base model for subsequent RL cold-start training.

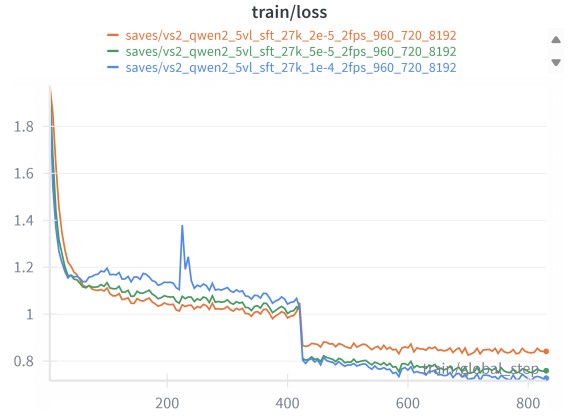

Figure 10: Training loss in ablations of learning rate, 2e-5, 5e-5, and 1e-4 are shown.

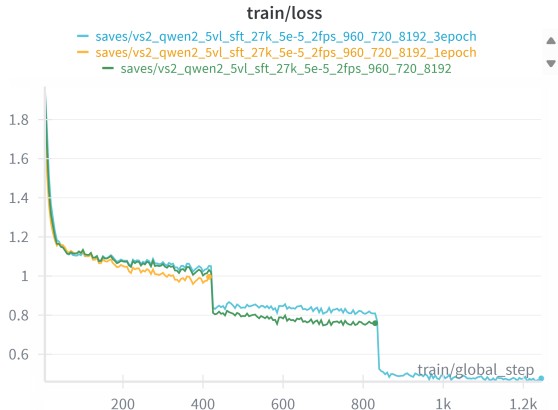

Figure 11: Training loss in ablations of training epoch, 1epoch, 2epoch, and 3epoch are shown.

Table 20: Ablation results on VIDEOSCORE-BENCH-V2 for different learning rate and epochs in SFT.

| SFT ablations | Accuracy | | | | Relaxed Accuracy | | | | PLCC | | | |
|---|---|---|---|---|---|---|---|---|---|---|---|---|
| | Visual | Align | Phy | Avg | Visual | Align | Phy | Avg | Visual | Align | Phy | Avg |
| Main (LR = 5e-5, 2epoch) | 43.69 | 40.88 | 34.87 | 39.81 | 90.38 | 86.97 | 83.77 | 87.04 | 56.74 | 58.24 | 44.72 | 53.23 |
| Ablation (LR = 1e-5) | 41.60 | 38.20 | 31.20 | 37.00 | 87.80 | 81.40 | 79.40 | 82.87 | 47.37 | 45.80 | 35.40 | 42.86 |
| Ablation (LR = 2e-5) | 42.77 | 38.55 | 34.94 | 38.75 | 90.76 | 85.14 | 80.72 | 85.54 | 54.17 | 52.77 | 40.99 | 49.31 |
| Ablation (LR = 1e-4) | 41.08 | 41.48 | 37.48 | 40.01 | 88.58 | 87.38 | 81.76 | 85.91 | 53.73 | 56.94 | 42.87 | 51.18 |
| Ablation (LR = 2e-4) | 41.48 | 40.48 | 37.48 | 39.81 | 89.38 | 87.58 | 83.37 | 86.78 | 51.93 | 56.15 | 45.53 | 51.20 |
| Ablation (1epoch) | 42.29 | 40.28 | 30.26 | 37.61 | 90.78 | 87.98 | 79.16 | 85.97 | 50.42 | 56.30 | 32.92 | 46.55 |
| Ablation (3epoch) | 45.29 | 37.28 | 38.88 | 40.48 | 92.39 | 87.38 | 85.77 | 88.51 | 58.34 | 56.71 | 49.60 | 54.88 |

## D.4 Ablation on inference settings

We also conduct an ablation on inference sampling rates, testing 2 fps, 4 fps, and 8 fps on VIDEOSCORE-BENCH-v2. Results in Table 21 show that 2 fps achieves the best performance, which aligns with our expectation: two frames per second are sufficient to capture most quality issues for evaluation, while higher frame rates introduce redundant information and potential noise that may interfere with the model's judgment.

Table 21: Ablation results on VIDEOSCORE-BENCH-v2 for different inference configurations.

| Inference | Accuracy | | | | Relaxed Accuracy | | | | PLCC | | | |
|---|---|---|---|---|---|---|---|---|---|---|---|---|
| Sampling fps | Visual | Align | Phy | Avg | Visual | Align | Phy | Avg | Visual | Align | Phy | Avg |
| 2fps | **50.10** | 43.88 | **39.08** | **44.35** | 92.99 | **91.38** | **87.98** | **90.78** | **60.13** | **62.60** | **52.73** | **60.37** |
| 4fps | 46.80 | **44.20** | 38.28 | 43.09 | 90.00 | 87.60 | 84.20 | 87.27 | 60.13 | 57.53 | 43.55 | 53.74 |
| 8fps | 41.67 | 40.77 | 37.61 | 40.02 | 85.81 | 88.96 | 83.78 | 86.18 | 56.28 | 58.27 | 41.86 | 52.14 |

## D.5 Ablation on RL training steps

We evaluated multiple intermediate checkpoints during RL training. Considering three evaluation metrics jointly, performance peaked around 300 steps. Beyond this point, scores on VIDEOSCORE-BENCH-v2 showed a clear decline., as shown in Table 22. Therefore, for all main experiments, we report results based on the 300-step checkpoint.

Table 22: Ablation on RL training steps. Accuracy and correlation between model answer and human score on VIDEOSCORE-BENCH-v2. *Relaxed Accuracy* counts cases where the prediction differs from the ground truth by at most one point.

| RL steps | Accuracy | | | | Relaxed Accuracy | | | | PLCC | | | |
|---|---|---|---|---|---|---|---|---|---|---|---|---|
| | Visual | Align | Phy | Avg | Visual | Align | Phy | Avg | Visual | Align | Phy | Avg |
| 200 | 50.50 | 42.89 | 39.28 | 44.22 | 92.79 | 91.59 | 87.80 | 90.73 | 65.14 | 62.95 | 57.60 | 61.90 |
| 300 | 50.10 | 43.88 | 39.08 | 44.35 | 92.99 | 91.38 | 87.98 | 90.78 | 65.78 | 62.60 | 52.73 | 60.37 |
| 400 | 46.20 | 43.80 | 36.00 | 42.00 | 92.80 | 90.20 | 85.40 | 89.47 | 64.57 | 58.87 | 44.61 | 61.72 |
| 500 | 47.60 | 45.80 | 40.00 | 44.47 | 90.20 | 91.40 | 87.80 | 89.80 | 61.57 | 60.59 | 52.10 | 58.09 |
| 600 | 50.60 | 43.40 | 41.40 | 45.13 | 91.40 | 89.90 | 87.20 | 89.50 | 62.89 | 56.49 | 51.62 | 57.00 |
| 700 | 48.00 | 45.20 | 38.00 | 43.73 | 90.80 | 88.20 | 87.60 | 88.87 | 64.28 | 57.07 | 49.43 | 56.93 |
| 833 | 45.00 | 45.60 | 37.80 | 42.80 | 91.40 | 89.60 | 85.60 | 88.87 | 64.68 | 59.19 | 46.04 | 56.64 |

### D.6 Discussion on CoT-like rationale in SFT

For preference benchmarks, we derive pairwise preferences from the sum of the three VideoScore2 dimension scores. To better understand why SFT without CoT can outperform SFT with CoT on OOD preference benchmarks (as shown in Table 7 in Section 4.6), we categorize incorrect predictions into two types:

- Cat I, where ground truth indicates no preference but the model predicts a preference for one video;

- Cat II, where the ground truth favors video A or B but the model predicts the wrong direction.

We also report the proportion of Cat II errors whose predicted score difference exceeds 3, which indicates high-confidence wrong preferences.

| Error Type | VGR-CoT | VGR-w/o CoT | T2VDB-CoT | T2VDB-w/o CoT |
|---|---|---|---|---|
| **Cat I** | 32.7% | 37.1% | 8.3% | 10.9% |
| **Cat II** | 67.3% | 62.9% | 91.7% | 89.1% |
| $\hookrightarrow \Delta > 3$ | 14.4% | 6.8% | 19.8% | 8.9% |

Table 23: Error-type breakdown of pairwise preference predictions for CoT ablations in SFT.

As shown in Table 23, CoT reduces Cat I errors on both benchmarks, suggesting that rationale-based SFT is more conservative when the two videos are close in quality. However, among wrong directional predictions (Cat II), CoT produces substantially more high-confidence errors: the fraction of $\Delta > 3$ errors is roughly doubled compared with SFT without CoT on both benchmarks. This suggests that CoT rationales can introduce a *reasoning-anchoring effect* under OOD preference evaluation: the model may anchor on plausible reasoning patterns learned from in-domain data and assign an overconfident preference in the wrong direction. In contrast, the score-only variant is less exposed to this anchoring effect, which helps explain its slightly better performance on preference benchmarks. This is a limitation of CoT-based SFT for OOD pairwise discriminability, although CoT remains beneficial for point-score benchmarks and for interpretability.

# E Case Studies

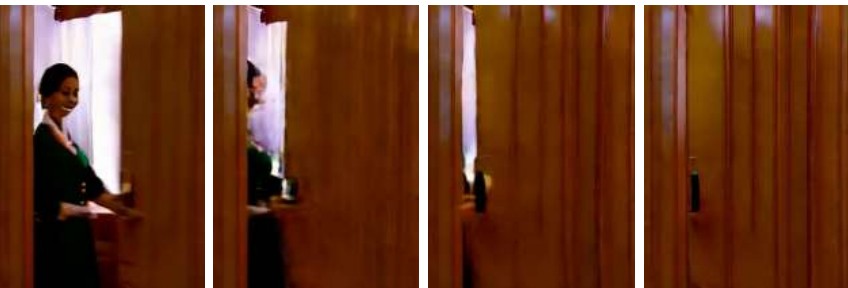

t2v-prompt: *A woman in Indian attire greeting in Indian style. The camera zooms out, and the doors shut widely in front of camera.*

VideoScore2:
<think>... Let me analyze this video…
**Visual Quality**:…visuals are severely degraded…, … frames exhibit extreme pixelation and heavy compression artifacts… Overall, the visual quality is critically compromised, justifying a very low rating (1/5).
**Text Alignment**: … camera does not perform a zoom-out, … no evident door-closing action…, The alignment with the intended prompt is only partially satisfied, yielding a moderate misalignment. (2/5)
**Physical Consistency**: severe blur and degradation obscure arm positions…with limited reliability… (2/5)
</think>
**Overall assess**: **1  2  2**

Figure 12: Case study of VIDEOSCORE2 evaluation (1)

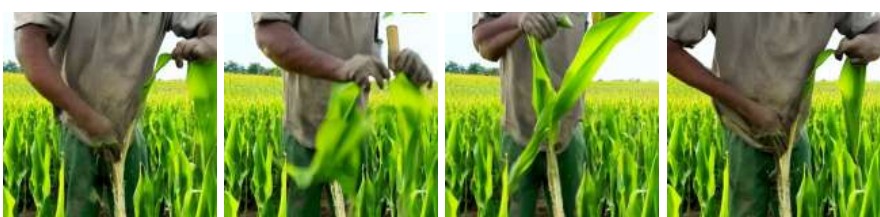

t2v-prompt: *A farmer in the field planting corn with a hoe, and the corn quickly matures. The farmer wipes off sweat and smiles.*

VideoScore2:
<think>... Let me analyze this video…
**Visual Quality**:…resolution is moderate; not high-end but acceptable …, Lighting is natural, … noticeable motion blur in certain moments…, Overall visual quality sits in the mid-range (3/5)
**Text Alignment**: …no rapid corn maturation is shown, no clear moment of wiping sweat or smiling… The alignment with the prompt is partial at best; core narrative elements are missing, …(2/5)
**Physical Consistency**: Some hand movements show anatomical inconsistencies…, Overall, physical plausibility is maintained with moderate inconsistencies. (3/5)
</think>
**Overall assess**: **3  2  3**

Figure 13: Case study of VIDEOSCORE2 evaluation (2)

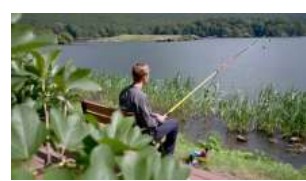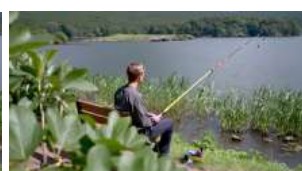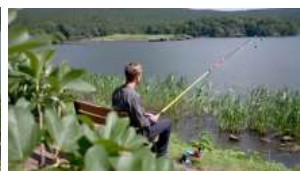

t2v-prompt: *A young blond man sitting on a bench, holding a fishing rod, with the fishing line out. Wide-angle.*

VideoScore2:
<think>... Let me analyze this video…
**Visual Quality**:…The resolution is very high and maintains crisp detail across scenes…, Colors are well-balanced,…wide-angle framing effectively captures the landscape…, Overall visual quality is excellent (5/5)
**Text Alignment**: …Holding a fishing rod – present…, Fishing line out – present and clearly visible…, alignment is essentially perfect (5/5)
**Physical Consistency**: The man's posture and grip look natural for fishing…, with lighting and shadows consistent with outdoor conditions. (5/5)
</think>
**Overall assess**: **5  5  5**

Figure 14: Case study of VIDEOSCORE2 evaluation (3)

## F   The Use of Large Language Models

Large language models (LLMs), including GPT-5, Gemini-2.5-Pro were used in the preparation of this paper. Their role was limited to supporting writing by suggesting phrasing alternatives, correcting grammar, and improving readability. All technical content, experimental design, analysis, and conclusions were created and verified by the authors.

