# OpenReview forum: "VideoScore2: Think Before You Score In Generated Video Evaluation"
_TMLR — Decision pending for TMLR_

### Review · Reviewer_N4Rg · 2026-03-04

**Summary Of Contributions:**

This paper presents VideoScore2, a multi-dimensional video quality evaluator for AI-generated content. The key contributions are: (1) a large-scale annotated dataset, VideoFeedback2, containing 27,168 human-annotated videos with scores and reasoning traces across three evaluation dimensions (visual quality, text alignment, and physical/common-sense consistency); (2) a two-stage training pipeline combining supervised fine-tuning (SFT) with reinforcement learning via GRPO to elicit chain-of-thought reasoning before scoring; and (3) empirical demonstration that the resulting model outperforms existing baselines on both in-domain and out-of-domain benchmarks, while also serving as a reward model for Best-of-N sampling in video generation.

**Audience:**

Yes

**Audience Explanation:**

Video generation evaluation is a timely and practically important problem. The combination of interpretable chain-of-thought reasoning with multi-dimensional scoring addresses a genuine gap in the literature, and the dataset itself (27,168 annotated videos) is a substantial community resource.

**Broader Impact Concerns:**

The paper includes an Ethics Statement that covers NSFW filtering, absence of PII, and annotator training. These are adequate for the core contributions. However, two points merit brief expansion:
Potential misuse of reward models. An interpretable, high-performing video reward model could be used to optimize generated video content in ways that score well on the three dimensions while potentially gaming the metric rather than producing genuinely high-quality or safe content. A brief discussion of this Goodhart's Law-type concern, particularly in the context of RL-based video generation training, would strengthen the ethics statement.
Annotator compensation and working conditions. The paper employs 15 annotators but does not disclose compensation, geographic location, or working conditions. Given growing awareness of annotation labor issues in the community, including this information would be appropriate.

**Claims And Evidence:**

Yes

**Claims Explanation:**

The paper provides quantitative results across multiple benchmarks (Tables 5 and 6) and conducts ablation studies covering training fps, learning rate, RL steps, score format, and CoT usage. The reported improvements are nontrivial (+5.94 average accuracy on the in-domain benchmark, +4.32 on OOD average).

**Requested Changes:**

1. Clarify in-domain vs. out-of-domain evaluation more carefully.
The test split VideoScore-Bench-v2 is carved from the same VideoFeedback2 pipeline. The paper should provide a more explicit discussion of what "in-domain" performance truly measures here, and whether the improvement over baselines reflects genuine quality estimation ability or adaptation to the annotation style, score distribution, and rationale vocabulary of the dataset. A cross-evaluator agreement analysis — e.g., how well baselines trained on completely disjoint data compare — would strengthen the validity of in-domain claims.
2. Address the circular evaluation concern in rationale generation.
The paper uses Claude-4-Sonnet (with thinking enabled) to expand human comments into full rationales, and then evaluates VideoScore2 (trained on these rationales) alongside Claude-4-Sonnet as a baseline in Table 5. This creates a potential circularity: the model is trained to mimic Claude's reasoning style on the same prompts, and then compared directly to Claude. This should be explicitly acknowledged, and the comparison with Claude-4-Sonnet should be interpreted in light of it. It may be appropriate to exclude Claude-4-Sonnet from the primary comparison table or add a dedicated discussion.
3. Inconsistent notation and possible data leakage in Table 4.
Table 4 reports statistics over "4,951 videos" but the total dataset is 27,168 videos. It is unclear whether this is a random subset, a specific annotation round, or a representative sample. This should be clarified. Furthermore, the 337 re-scored entries that were discarded represent roughly 6.8% of this subset — the criteria for discarding (after up to three re-scoring attempts) should be stated precisely.
4. Best-of-N experimental design needs strengthening.
The current Best-of-N setup uses VBench as the downstream quality measure, but VBench uses its own internal metrics that do not necessarily align with VideoScore2's three dimensions. The paper should either (a) justify why VBench dimensions are an appropriate proxy, or (b) use human preference studies as the downstream signal. Without this, it is difficult to conclude that VideoScore2 is a reliable reward model rather than one that happens to correlate with VBench's automatic metrics.
5. Ablation Table 7 (RL w/ vs. w/o SFT) uses different row labels than the text.
The text refers to "RL w/ SFT (44.53)" in Table 7 but the main results in Table 5 report "44.35" for VideoScore2. This discrepancy should be resolved and explained. Similarly, the "w/ CoT" row shows 39.81 accuracy, which is labeled as the SFT-only result, but Section 4.6 discusses this in the context of RL — the table structure is confusing and should be reorganized for clarity.
6. Broader comparison to prompting-based evaluators.
Gemini-2.5-Pro and GPT-5 are included as baselines but are prompted in zero-shot fashion. Given that these models are significantly larger and more general than the 7B VideoScore2, the comparison is informative but somewhat asymmetric. A few-shot prompting condition or chain-of-thought prompting for the VLM baselines would provide a fairer comparison and better isolate the benefit of fine-tuning versus prompting.

---

> ### Author Response · Authors · 2026-04-09
> **Author's Response to the Review (1/2)**
>
> > ### RC1: in-domain v.s. out-of-domain evaluation
>
> We agree that in-domain performance reflects both genuine quality estimation ability and adaptation to the annotation protocol, as is common across the literature. However, the improvements are not purely due to stylistic adaptation: (1) training and test sets contain non-overlapping videos from different models and prompts, preventing memorization; (2) all baselines receive identical instructions, so the gap cannot be attributed to VideoScore2 merely knowing the annotation format; (3) our OOD results on benchmarks with entirely different annotation pipelines demonstrate transferable evaluation ability beyond dataset-specific patterns.
>
> Regarding cross-evaluator agreement, our OOD evaluation effectively serves a similar purpose by testing VideoScore2 against benchmarks annotated by independent teams. We will add an explicit discussion of what in-domain evaluation captures and its limitations
>
> > ### RC2: circular evaluation concern
>
> We acknowledge the potential circularity and would like to clarify:
>
> (1) Claude-4-Sonnet was used to **expand** existing human annotations (scores and short comments) into structured rationales, it did not independently generate the core judgments. (2) For scores: as described in Section 3.3, when the human–model difference is 0 or 1, the **human score is used directly**; only when the difference equals 2 (~**15%** of cases) is the average taken. Claude's influence on training labels is therefore bounded, affecting a small portion of data with at most a 1-point shift.
>
> In the revised version, we will (1) mark Claude-4-Sonnet's dual role in Table 5, and (2) add a limitations discussion noting that VideoScore2's rationale style may inherit characteristics from Claude and that ~15% of training scores are partially influenced by Claude's predictions.
>
> > ### RC3: concerns with Table4
>
> **On the 4,951 videos**: This subset corresponds to ~20% of each annotator's assigned videos, selected for quality verification as described in Section 3.2. These are videos with both human and model-assisted scores available for comparison.
>
> **On the Table 4 statistics**: The upper section treats all three dimension scores as a flattened list (4,951 × 3 = 14,853 scores). The absolute differences between human and model scores are distributed as: 0 (4,710), 1 (7,698), 2 (2,062), ≥3 (383). Note that ≥3 cases are **residual** differences that persist even after up to three rounds of re-scoring.
>
> **On the discard criteria**: If any video has a difference ≥3 on at least one dimension, it is sent for re-scoring (up to three attempts). If the difference still remains ≥3 after three attempts, the video is discarded. This per-video criterion explains why 337 videos (6.8%) were discarded despite only 383 individual scores (2.6%) having ≥3 differences, a single problematic dimension is sufficient to discard the entire video.

---

> ### Author Response · Authors · 2026-04-09
> **Author's Response to the Review (2/2)**
>
> > ### RC4: Best-of-N results
>
> We thank the reviewer for this point. We do not use all VBench dimensions:
> as described in Appendix C.4, we selected a subset (subject consistency, background consistency, aesthetic quality, imaging quality, and motion smoothness) that are closely aligned with VideoScore2's visual quality and physics consistency dimensions. Dimensions lacking clear correspondence (e.g., class consistency, multiple objects) were excluded.
>
> We chose VBench as the downstream signal because it is the most widely adopted automatic benchmark in video generation, enabling direct and reproducible comparison with other reward models under a standardized protocol. We agree that human preference studies would provide stronger validation and consider this valuable future work, though conducting reliable large-scale human evaluation introduces challenges in cost and reproducibility. We will revise the paper to better justify our VBench dimension selection.
>
>
>
> > ### RC5:
>
> We thank the reviewer for the careful reading. The value "44.53" in Table 7 is indeed a typo, the correct number should be **44.35**, consistent with Table 5. We apologize for the confusion and will correct this in the revised version.
>
> Regarding the table structure: in Table 7, the ablations are organized into three groups. The first group (RL w/ SFT vs. RL w/o SFT) ablates the effect of SFT initialization. The second group (w/ CoT vs. w/o CoT) compares **SFT-only** models with and without chain-of-thought rationales, this comparison is independent of RL and is discussed as such in Section 4.6. The third group ablates the score normalization strategy. We acknowledge that this grouping is not sufficiently clear in the current presentation, and the row labeled "RL w/ SFT" in the first group and "Normalized Score (default)" in the third group both correspond to the final VideoScore2 model reported in Table 5.
>
> We will also reorganize Table 7 in the revised version with explicit group headers and consistent labeling to improve clarity.
>
> > ### RC6: few-shot results of MLLM
> We conduct additional experiments with 3-shot prompting for all VLM baselines. The results are shown in the table below:
>
> | Model | Setting | Acc-Vis | Acc-T2V | Acc-Phy | Acc-Avg | RelaxAcc-Vis | RelaxAcc-T2V | RelaxAcc-Phy | RelaxAcc-Avg | PLCC-Vis | PLCC-T2V | PLCC-Phy | PLCC-Avg |
> |---|---|---|---|---|---|---|---|---|---|---|---|---|---|
> | GPT-5 | 0-shot | 30.72 | 27.91 | 20.08 | 26.24 | 73.90 | 72.29 | 60.84 | 69.01 | 13.38 | 23.34 | 17.24 | 17.99 |
> | GPT-5 | 3-shot | 37.90 | 35.28 | 25.61 | 32.93 | 81.25 | 76.41 | 70.16 | 75.94 | 38.27 | 43.51 | 34.39 | 38.72 |
> | Gemini-2.5-Pro | 0-shot | 29.92 | 29.72 | 24.07 | 27.90 | 71.49 | 70.88 | 61.45 | 67.94 | 26.71 | 32.96 | 19.75 | 26.47 |
> | Gemini-2.5-Pro | 3-shot | 29.03 | 30.44 | 22.58 | 27.35 | 74.60 | 76.82 | 63.71 | 71.71 | 50.24 | 50.65 | 45.66 | 45.66 |
> | GLM-4.1v | 0-shot | 33.27 | 31.46 | 21.42 | 28.72 | 80.76 | 77.15 | 61.22 | 73.04 | 28.03 | 16.80 | 10.18 | 18.34 |
> | GLM-4.1v | 3-shot | 34.62 | 31.57 | 25.26 | 30.48 | 81.47 | 80.86 | 66.80 | 76.38 | 30.99 | 35.03 | 16.36 | 27.46 |
>
> Two observations can be drawn from these results. Firstly, few-shot prompting does yield improvements across most metrics and models, confirming that in-context examples help VLMs better understand the video quality assessment task. Notably, GPT-5 benefits the most, with its average PLCC improving from 17.99 to 38.72. Second, despite these gains, a substantial performance gap remains between the prompted VLMs and VideoScore2, even when the VLMs are significantly larger in scale. This suggests that few-shot prompting alone cannot close the gap with task-specific fine-tuning, and reinforces the value of our fine-tuning approach on a dedicated video quality dataset.
>
> We will include them in the revised version to provide a more comprehensive comparison.

---

### Review · Reviewer_rrZZ · 2026-03-09

**Summary Of Contributions:**

This paper presents VideoScore2, an interpretable evaluator for AI-generated videos that produces chain-of-thought rationales before scoring along three dimensions (visual quality, text alignment, physical/common-sense consistency). The main contributions include:

* An evaluator that combines multi-dimensional scoring with explicit reasoning, trained with a two-stage SFT+RL pipeline. Experiments show it achieves higher agreement with human judgments than existing methods on both in-domain and out-of-domain benchmarks.
* A large-scale dataset VideoFeedback2 (27,168 videos from 22 T2V models) with human-annotated scores and LLM-elicited reasoning traces to support training.

Strengths:

* The interpretability angle is well-motivated. Existing reward models output single opaque scores; providing structured reasoning alongside multi-dimensional scores is a clear step forward for debugging and understanding video generation quality.
* The dataset covers a wide quality range by sampling from four model tiers per prompt, and includes manually designed prompts for known T2V weaknesses (multi-action, OCR, camera motion). The score alignment between human annotators and LLM rationales handles disagreements reasonably.
* The evaluation covers in-domain and four OOD benchmarks of two types (preference and point-score).

weakness:
* The rationales are not human-written. They are generated by Claude-4-Sonnet from brief human comments, then aligned by GPT-5-mini. The paper claims the model learns "how to reason," but it learns to mimic LLM-style reasoning, not human evaluator reasoning. No independent evaluation of rationale quality (faithfulness, accuracy, hallucination rate) is provided.
* The CoT ablation (Table 7) shows rationales hurt OOD preference performance. The paper explains this as "more conservative scores increasing tie rate," but this undermines the core claim.

**Audience:**

Yes

**Audience Explanation:**

Yes. The question of how to make video evaluation interpretable and multi-dimensional is increasingly relevant as T2V models improve. The dataset construction pipeline, the SFT+RL training recipe, and the finding that rationales help point-score but hurt preference tasks are all useful to the community. The analysis of where current evaluators still struggle (physics, common sense) points to important open problems.

**Broader Impact Concerns:**

No major concerns. The work focuses on evaluation rather than generation. Standard considerations about potential misuse of video quality metrics for optimizing misleading content apply but are not specific to this paper.

**Claims And Evidence:**

Yes

**Claims Explanation:**

In-domain results are solid and the two-stage training ablations are convincing. Two concerns: (1) the OOD superiority claim relies on averaging across 4 benchmarks where the model underperforms on several individual ones (e.g., VideoGenReward-Bench: 51.53 vs VideoReward's 59.69); (2) rationale quality—the paper's core differentiator—is never independently evaluated for faithfulness or accuracy.

**Requested Changes:**

Major:

* Provide human evaluation of rationale quality (e.g., sample 100+ rationales, have annotators rate faithfulness and accuracy).
* Discuss OOD results per-benchmark rather than relying on overall average. Acknowledge where VideoScore2 underperforms  and analyze why—is it the CoT conservatism, dimension mismatch, or something else.

Recommended:
* Report inference cost (time, tokens) compared to score-only models

---

> ### Author Response · Authors · 2026-04-09
> **Author's Response to the Review**
>
> > ### RC1: verify rationale quality manually
>
> We thank the reviewer for this constructive suggestion. We fully agree that independent human evaluation of rationale quality is necessary. We sampled 200 rationales and had human annotators evaluate them on two aspects: (1) **Faithfulness**: whether the rationale accurately reflects the content of the original annotated short comment, and (2) **Correctness**:  whether rationale is consistent with video content. Each aspect was rated on a three-level scale, good, moderate, and bad:
>
> | Aspect | Good | Moderate | Bad |
> |---|---|---|---|
> | Faithfulness | 74.50% | 23.00% | 2.50% |
> | Correctness | 81.50% | 16.50% | 2.00% |
>
> The results show the vast majority of rationales are correct with respect to video content (81.5% Good, only 2% Bad). For faithfulness, the Moderate cases (23%) primarily involve **partial omission** of details mentioned in human comments. For example, missing background-related issues or specific momentary action artifacts. Only 2.5% of rationales contain descriptions that conflict with the original human comments. The rationales capture the substantive content of human judgments. We will include this human evaluation in the revised version along with a discussion of the rationale generation pipeline's limitations.
>
> > ### RC2: more detailed analysis on OOD Benchmarks
>
> We thank the reviewer for this constructive suggestion. We will include per-benchmark analysis in the revised version.
>
> For preference benchmarks, we derive pairwise preferences from the sum of VideoScore2's three dimension scores. To better understand the error patterns, we categorize incorrect predictions into two types:
> - **Cat I (GT=same, pred. diff.)**: the ground truth indicates no preference, but prediction gives one video is better.
> - **Cat II (GT=A/B, pred. wrong)**: the ground truth favors video A or B, but prediction is wrong
>
> We further report the proportion of Cat II errors where the predicted score difference between the two videos exceeds 3, which indicates high-confidence but incorrect predictions.
>
> | Error Type | VGR-CoT | VGR-w/o | T2VDB-CoT | T2VDB-w/o |
> |---|---|---|---|---|
> | Cat I | 32.7% | 37.1% | 8.3% | 10.9% |
> | Cat II | 67.3% | 62.9% | 91.7% | 89.1% |
> | └  Δ>3 | 14.4% | 6.8% | 19.8% | 8.9% |
>
> We found that our original explanation of "conservative scores increasing tie rate" is incomplete. The data reveals a more nuanced pattern consistent across both benchmarks:
>
> (1) **CoT helps more accurate preferences on same-quality pairs** (lower Cat I rate), confirming a beneficial conservative tendency when videos are close in quality.
>
> (2) **However, CoT significantly increases high-confidence errors**: when CoT predicts wrong preference direction, it does so with larger score gaps (Δ>3 errors roughly double compared to w/o CoT on both benchmarks). This indicates CoT does not simply compress scores, it introduces **miscalibration** in OOD settings, where reasoning process can anchor on features learned from in-domain data that do not transfer well.
>
> We believe this reflects a reasoning-anchoring effect: CoT rationales trained on in-domain patterns may construct plausible-sounding but incorrect justifications for OOD content, leading to overconfident misjudgments. The score-only model, lacking explicit reasoning, is less susceptible to such anchoring. We acknowledge this as a genuine limitation. The primary value of CoT lies in (a) improved in-domain accuracy and (b) interpretability for diagnostic purposes.
>
>
> > ### RC3: inference cost
>
> We report the average per-video inference cost across 500 videos, measured on a single NVIDIA A6000 (48GB) GPU.
>
> | Method | Inference Time (s) | Tokens (Input + Output) |
> |---|---|---|
> | VideoScore2 | 24.1 | 1605 + 418 |
> | VideoReward | 1.6 | 1610 + 0 |
> | VisionReward | 10.3 | 8411 + 58 |
> | UnifiedReward | 2.7 | 2447 + 34 |
> | VideoPhy2-AutoEval | 4.1 | 355 + 8 |
> | ImageReward (2fps, max 16 frames) | 41.3 | 3792 + 2 |
>
> VideoScore2 has higher inference time than other methods, which is expected: in addition to scalar scores, it generates detailed rationales as interpretable evaluation outputs. When measured in terms of time per token, VideoScore2 is more efficient than the baselines.
>
> Several notes on the token differences:
>
> VideoReward reports 0 output tokens because its score is produced by a linear regression head rather than text generation.
>
> VisionReward has significantly higher input tokens because it performs multiple independent binary evaluations per dimension and aggregates the results into a composite score.
>
> UnifiedReward has higher input tokens because it is based on LLaVA, whose CLIP/ViT image encoder produces 80-100 more tokens per frame than Qwen2.5-VL's native vision encoder. Its chat template also introduces additional overhead. ImageReward also use a CLIP/ViT encoder.
>
> VideoPhy2-AutoEval has lower input tokens due to lower image encoding resolution and shorter evaluation prompts in original implementation.

---

### Review · Reviewer_88Sb · 2026-03-18

**Summary Of Contributions:**

The paper introduces a multi-axis framework for evaluating text-to-video models along three dimensions: visual quality, text-video alignment, and physical/common-sense consistency.
To support this, the authors collect a large-scale benchmark of human annotations, including both scores and associated reasoning traces.
Based on this dataset, they train a scoring model using a two-stage optimization procedure consisting of supervised fine-tuning followed by reinforcement learning.
The resulting model achieves strong performance on both the proposed benchmark and several out-of-domain benchmarks.

**Strengths**

- The paper is clearly written and provides detailed descriptions of the data collection, annotation protocol, and scoring model training pipeline.

- The proposed large-scale benchmark has the potential to be a valuable resource for the community.

- The experimental section is thorough, with extensive ablations that improve understanding of the method and support reproducibility.

**Weaknesses**

1. It is unclear how chain-of-thought reasoning contributes to evaluating video models or to its use in a reward modeling setup. In practice, video generation and evaluation are typically performed at scale, and it is not evident how detailed score justifications would translate into improved model quality during post-training. Clarifying the practical benefit of these reasoning traces would strengthen the paper.

2. Multi-dimensional evaluation for generative models is not entirely new, as prior work such as VisionReward has explored similar directions. The paper would benefit from a clearer positioning of its contributions relative to this line of work.

3. The use of LLMs to expand or refine annotator-provided rationales is concerning. Such models can introduce hallucinations and may reduce the diversity of reasoning traces, potentially biasing the dataset. Some validation or analysis of this step would be helpful.

4. The related work on vision reward models in Section 2.2 is quite limited and omits several relevant works particularly from the image  domain (e.g., [1–6]). Expanding this section would improve coverage and positioning.

5. The choice of evaluation dimensions and challenges (e.g., multi-action, OCR, camera motion) is not fully justified. Other important aspects such as human poses, object dynamics (such as vehicles or bicycles), or dense scenes are not considered, which may bias the scoring model toward a subset of challenges. A clearer motivation for this taxonomy is needed.

6. The statement that annotators with inconsistent work received feedback raises concerns about potential bias. This process may implicitly enforce inter-annotator agreement, reducing diversity in judgments and effectively optimizing for agreement rather than capturing genuine variability. Clarification on how this is controlled would be useful.

[1] VisionReward: Fine-Grained Multi-Dimensional Human Preference Learning for Image and Video Generation

[2] Human Preference Score v2: A Solid Benchmark for Evaluating Human Preferences of Text-to-Image Synthesis"

[3] HPSv3: Towards Wide-Spectrum Human Preference Score

[4] FineGRAIN: Evaluating Failure Modes of Text-to-Image Models with Vision Language Model Judges

[5] MMIG-Bench: Towards Comprehensive and Explainable Evaluation of Multi-Modal Image Generation Models [6] Rich Human Feedback for Text-to-Image Generation

**Audience:**

Yes

**Audience Explanation:**

Yes, the paper is likely to be of interest to a subset of the TMLR audience, particularly researchers working on evaluating generative models. While it does not present fundamentally new findings or breakthroughs, it includes extensive experiments and ablations that can provide
useful insights for designing and training scoring models.

**Broader Impact Concerns:**

The authors provided an ethical statement which covers potential implications reasonably well.

**Claims And Evidence:**

Yes

**Claims Explanation:**

The paper's general claims are supported by clear empirical evidence. The paper includes a well-described benchmark, a detailed annotation pipeline, and extensive experiments with ablations, which make the results convincing within the scope of the proposed setup. The reported improvements on both the proposed and out-of-domain benchmarks further support the effectiveness of the approach.

However, some claims would benefit from stronger justification. In particular, the practical value of reasoning traces for evaluation and reward modeling is not clearly demonstrated, and the use of LLMs to refine rationales raises concerns about potential bias and reduced diversity. Additionally, the positioning with respect to prior work on multi-dimensional evaluation is not fully clarified. Strengthening these aspects would make the evidence more comprehensive.

**Requested Changes:**

- Clarify the role and practical benefit of reasoning traces, particularly in the context of large-scale evaluation and reward modeling. It should be demonstrated how these traces improve model quality, training stability, or interpretability.
- Better position the work with respect to prior multi-dimensional evaluation frameworks (e.g., VisionReward) and clearly articulate the novelty of the proposed approach.
- Provide justification for the chosen evaluation taxonomy (e.g., multi-action, OCR, camera motion) and discuss its coverage. It would strengthen the paper to explain why these dimensions are sufficient and how missing categories might affect the scoring model.
- Address concerns regarding the use of LLMs to expand or refine annotator rationales.
- Clarify the annotator feedback process and its potential impact on bias and inter-annotator agreement.
- Expand the related work section on reward models to include additional relevant literature (e.g., [1–6]).

---

> ### Author Response · Authors · 2026-04-09
> **Author's Response to the Review**
>
> > ### RC1: role of reasoning traces
>
> We thank the reviewer for this important point. The reasoning traces serve two purposes:
>
> (1) **Diagnostics beyond scalar scores**: Two videos may both receive a low physics score but for very different reasons, e.g., unnatural object trajectories vs. incorrect fluid dynamics. Rationales expose these fine-grained distinctions, enabling developers to pinpoint failure modes, which is not possible from scalar scores alone.
>
> (2) **Improved score prediction**: The rationales serve as intermediate supervision that helps the model better calibrate its predictions. As shown in Table 7, the CoT variant achieves higher in-domain accuracy, though we acknowledge a trade-off in OOD settings that warrants further investigation.
>
> > ### RC2: positioning
>
> We agree that multi-dimensional evaluation is not new, and prior works such as VisionReward have made valuable contributions in this direction. Our work builds upon this line of research and differs in two key aspects: (1) **Transparent rationales**: VideoScore2 generates structured reasoning alongside scores, enabling users to understand where and why quality issues arise, rather than providing only scalar scores. (2) **Human-supervised annotation at scale**: our dataset is constructed with human verification, resulting in stronger alignment with human perception compared to approaches relying primarily on synthetic data. These are complementary contributions, we advance the existing paradigm by adding interpretability and stronger human grounding.
>
> > ### RC3: evaluation subset like text-OCR
>
> We would like to clarify that these subsets are **not intended as a comprehensive taxonomy**, but rather as **representative challenging scenarios** selected based on failure modes discussed in previous works and video generation community.
> Regarding the concern about potential bias in the scoring model: while our training set does include prompts from these specific challenging scenarios, they constitute **less than 25% of the total training data**. The majority of training prompts are sampled from **general-purpose prompt datasets** covering a wide range of content types and scenarios. Therefore, the scoring model is not disproportionately optimized for these particular challenges, and its training distribution remains broadly representative.
>
> To verify that these subsets indeed capture challenging scenarios, we compared the average annotation scores:
> | Set | Visual | T2V-Align | Physics | Overall |
> |---|---|---|---|---|
> | Full Set | 2.90 | 3.15 | 2.84 | 2.96 |
> | OCR/Text | 2.74 | 2.49 | 2.64 | 2.63 |
> | Story | 2.75 | 2.65 | 2.52 | 2.64 |
> | Camera Motion | 2.88 | 3.04 | 2.86 | 2.91 |
>
> We acknowledge that dimensions such as human pose accuracy and object dynamics are also important and can be incorporated into diagnosis in future work.
>
> > ### RC4: use of LLMs to expand the rationale
>
> Thank the reviewer for this concern. We share the awareness that LLM-generated expansions may introduce hallucinations or reduce reasoning diversity, and have taken steps to validate this.
>
> **On hallucination risk:** We conducted a human evaluation on 200 sampled rationales, assessing faithfulness to the original human comments and correctness against video content:
> | Aspect | Good | Moderate | Bad |
> |---|---|---|---|
> | Faithfulness | 74.50% | 23.00% | 2.50% |
> | Correctness | 81.50% | 16.50% | 2.00% |
>
> The Bad rate is low (2.5% for faithfulness, 2% for correctness), indicating that hallucination is not a prevalent issue. The Moderate cases in faithfulness are primarily partial omissions of details from human comments rather than fabricated content.
>
> **On diversity reduction:** We note that the LLM expansion is conditioned on **individual human annotator comments**. The LLM serves as a structured elaboration tool rather than a generator of independent reasoning, so the diversity of the underlying human judgments is largely preserved in the expanded rationales.
>
> > ### RC5: annotator feedback process
>
> We would like to clarify that the feedback provided during annotation trial rounds focused on correcting misunderstandings of the annotation guidelines, rather than enforcing agreement on particular videos. As reflected in Table 3, our agreement metric treats score differences within 1 point as consistent, acknowledging the inherent subjectivity of video quality assessment. We agree with the reviewer that preserving genuine variability in judgments is important, as video evaluation is an inherently subjective task, and we did not attempt to force annotators to follow a uniform score distribution.
>
> > ### RC6: expand related works
>
> Thanks for pointing out this. We will expand Section 2.2 in the revised version to include the suggested references [1–6] and provide a more thorough discussion of reward models in the image domain, with more clarification.
>
> > ### Summary
>
> We will revise our paper and clarify all the essantial points mentioned above in the revised version.

---

### Decision · Action_Editor_NuQj · 2026-06-13

**Recommendation:** Accept with minor revision

**Audience:**

Yes

**Audience Explanation:**

Video generation, and LLM/VLM post-training, is a burgeoning field within the TMLR community, and this paper will therefore be of interest to a substantial portion of the TMLR community.

**Claims And Evidence:**

Yes

**Claims Explanation:**

The authors present a video-quality / reward model for AI-generated videos. In contrast to prior works, the proposed model does not output a single score, but provides scores across multiple axes and also a textual rationale. The main contribution of this work is a large-scale dataset, VideoFeedback 2, which has about 27K human-annotated video scores, and reasoning traces which are expanded by Claude Sonnet. The authors also show how training a model with this dataset (first SFT for warm-starting, then RL with GRPO) leads to state-of-the-art results across a number of video evaluation benchmarks. The authors also show that the proposed model can be used for Best-of-N sampling to improve the samples from a video generation model.

The reviewers generally felt that the paper was well motivated, and that the claims were well substantiated with experimental results. However, a key concern by a number of the reviewers was about the reasoning traces: The reasoning traces are expanded by an LLM (ie Claude) from short rationales provided by human annotators. The AE understands that these reasoning traces are helpful to provide warm-start SFT data, and that subsequent RL from this initialisation improves performance. However, the reasoning traces themselves are not necessarily accurate, and should not be used for video evaluation. Moreover, as LLMs continue to evolve at a rapid rate, their actual utility in warm-starting newer LLMs is also questionable. Therefore, the AE feels that the claims around the rationales could be toned down, and that the paper stands by itself based on its other, well-substantiated contributions.